# CRAFTING HEAVY-TAILS IN WEIGHT MATRIX SPECTRUM WITHOUT GRADIENT NOISE

## ABSTRACT

Training strategies for modern deep neural networks (NNs) tend to induce a heavy-tailed (HT) empirical spectral density (ESD) in the layer weights. While previous efforts have shown that the HT phenomenon correlates with good generalization in large NNs, a theoretical explanation of its occurrence is still lacking. Especially, understanding the conditions which lead to this phenomenon can shed light on the interplay between generalization and weight spectra. Our work aims to bridge this gap by presenting a simple, rich setting to model the emergence of HT ESD. In particular, we present a theory-informed analysis for 'crafting' heavy tails in the ESD of two-layer NNs without any gradient noise. This is the first work to analyze a noise-free setting and incorporate optimizer (`GD`/`Adam`) dependent (large) learning rates into the HT ESD analysis. Our results highlight the role of learning rates on the Bulk+Spike and HT shape of the ESDs in the early phase of training, which can facilitate generalization in the two-layer NN. These observations shed light on the behavior of large-scale NNs, albeit in a much simpler setting. Last but not least, we present a novel perspective on the ESD evolution dynamics by analyzing the singular vectors of weight matrices and optimizer updates.

## 1 INTRODUCTION

By employing techniques from random matrix theory, Martin & Mahoney (2021a) observed an implicit self-regularization of the training process, due to which deep NN weights exhibit HT ESDs. Especially, they observed a correlation between the heaviness of tails in the ESDs and the strong generalization performance of deep NNs on CV/NLP tasks. On the other hand, owing to the HT nature of stochastic gradient noise (Simsekli et al., 2019; Panigrahi et al., 2019; Zhang et al., 2020), further studies have shown that limiting HT distributions of weight values lead to good generalization (Simsekli et al., 2020a; Gurbuzbalaban et al., 2021; Hodgkinson et al., 2022; Simsekli et al., 2019). Since such a distribution of values can, in turn, result in weight matrices with HT ESDs (Arous & Guionnet, 2008; Belinschi et al., 2009), the study of this broader HT phenomenon in deep learning has gained importance in recent years. In particular, the shape of HT ESDs has been successfully leveraged to assess the quality of pre-trained NNs (Martin & Mahoney, 2020; Martin et al., 2021; Martin & Mahoney, 2021b) (including Large Language Models (Yang et al., 2023)), and design layer-wise learning rate schedulers (Zhou et al., 2023). Similarly, the shape of HT stochastic noise has been shown to impact generalization (Simsekli et al., 2019; 2020b;a; Hodgkinson et al., 2022; Gurbuzbalaban et al., 2021), the compressibility of NNs (Barsbey et al., 2021), and the design of federated-learning algorithms (Yang et al., 2022; Li et al., 2024). The theoretical efforts (Simsekli et al., 2019; 2020a;b; Gurbuzbalaban et al., 2021; Hodgkinson & Mahoney, 2021; Hodgkinson et al., 2022) that aim to understand the underlying mechanisms of the HT phenomenon have focused on the limiting distributions of the weight values and have not explicitly studied the ESDs.

In this paper, we present an alternative setup that is amenable to discrete-step analysis and is rich enough to study the evolution of ESDs in two-layer NNs without any gradient noise. Our setup does not rely on continuous time approximations to SGD or limiting distribution analysis but instead employs a *Teacher-Student* setting (Arous et al., 2021; Bietti et al., 2022; Ba et al., 2022; Dandi et al., 2023; Ba et al., 2023; Mousavi-Hosseini et al., 2023b) to study the effects of finite optimizer steps on the weight matrix ESD (Figure 1). By training a two-layer feed-forward NN (*Student*) to learn a single-index model (*Teacher*) using vanilla Gradient Descent (`GD`) and Full-batch Adam

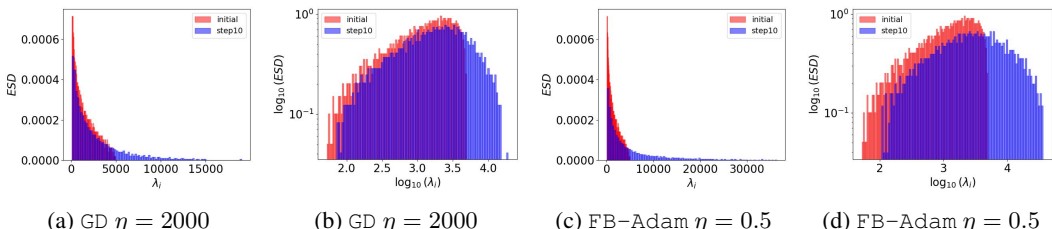

(a) GD $\eta = 2000$      (b) GD $\eta = 2000$      (c) FB-Adam $\eta = 0.5$      (d) FB-Adam $\eta = 0.5$

Figure 1: Emergence of HT spectra after 10 GD/FB-Adam steps. Both linear and log scales are shown. For the purpose of illustrating clear HT shapes, we did not choose a highly HT spectrum. Highly HT spectra can also be generated without gradient noise and are discussed in Appendix A.

(FB-Adam) optimizers, we show that: *"The ESD of the hidden layer weight matrix exhibits heavy tails after multiple steps of GD/FB-Adam with (sufficiently) large learning rates"*.

**Intuition.** Before diving into the details, we present the reader with a mental model of the ESD evolution in this setup. At initialization, the hidden layer weight matrix ESD of the student NN can be characterized by the Marchencko-Pastur distribution (i.e. random-like). The first step with a *large* $\eta$ has been shown to result in an outlier singular value (i.e. a 'spike') in the ESD and results in a 'Bulk+Spike' shape. The singular vector corresponding to this spike tends to align with the target direction of the teacher model and results in improved generalization (Ba et al., 2022; Dandi et al., 2023). By continuing to train the student NN with such *large* $\eta$, our work shows that the interactions between the spike and the bulk gradually lead to a 'Bulk-Decay' and finally lead to HT ESD.

Our theory is primarily aimed at identifying the *large* $\eta$ for FB-Adam, which can result in a spike after the one-step update. Especially, we formulate the scale of $\eta$ (depending on hidden layer width and data dimension) which can lead to a Bulk+Spike ESD after the one-step update. We also show their intriguing relationship with *feature-learning* (Ba et al., 2022; Dandi et al., 2023) in two-layer NNs, which can lead to improved generalization. Our results for FB-Adam in the feature learning context can be of independent interest to the community since prior works are limited to GD.

Toward understanding the shape transitions of ESD from the Bulk+Spike to an HT, we vary $\eta$ from small to large values and observe that the distance between the spike and the bulk (i.e spectral gap) determines the HT emergence during the early phases of training. For instance, $\eta = 1$ leads to HT ESDs just after $t = 10$ steps, whereas $\eta = 0.1$ requires very long training up to $t = 10000$ steps (see also Table 3, Table 4). Thus highlighting the necessity to study critical learning rates for one-step optimizer updates. We also analyze the correlations between the heaviness of tails (as measured by power-law fits) in the ESD and the generalization of the student NN after multiple steps. We showcase empirical results on the existence of a range of $\eta$ (depending on the optimizer) that can result in an HT ESD as well as improve the generalization of the student NN. In particular, $\eta$ in such a suitable range results in HT ESDs whose power-law fit (as measured by the hill-climbing approach (Yang et al., 2023)) lie in the range of $(2, 2.5)$ and lead to strong generalization. This result supports the observations of Martin & Mahoney (2021a); Martin et al. (2021) for well-trained deep NNs where the truncated power-law fit of the ESDs lies within a range of $(2, 4)$. Finally, we present a novel perspective on HT ESD emergence using tools from the 'signal recovery' literature (Landau et al., 2023; Benaych-Georges & Nadakuditi, 2012). In particular, we show that the alignments of singular vectors of the hidden layer weight matrix and its corresponding optimizer update matrix significantly influence the shape of ESD during training.

To summarize, our main contributions are as follows:

- We present a gradient-noise-free setting to study the emergence of HT ESD in the hidden layer weight matrix of two-layer NNs during the initial phases of training. To the best of our knowledge, this is the first work to study the early evolution phases of the ESD across discrete training steps with large learning rates. In this setup, we present a novel insight that feature learning after the first step (via the spike in the weight matrix ESD) facilitates HT emergence after finite steps.

- We theoretically establish the scale of the learning rate at which one step of FB-Adam results in a spike in the hidden layer weight matrix ESD. This result can be of independent interest to the community since prior results have been established only for GD.

- We empirically analyze the evolution of the hidden layer weight matrix ESD into an HT distribution during training. Interestingly, we show that for a certain range of optimizer-dependent learning rates, the two-layer NN can exhibit HT ESDs and generalize well.

- We present a novel perspective on the interactions between the bulk and spike of the ESD which result in a HT distribution. By analyzing the overlap(alignment) of singular vectors of the weight matrix and its corresponding optimizer updates during training, our qualitative observations show that the alignments tend to exhibit an HT-like decay across the singular vector indices and correlate with the emergence of HT ESDs during training.

Overall, our paper makes multiple novel theoretical and empirical contributions centered around the emergence of HT ESDs and their connection to good generalization of NNs.

## 2 RELATED WORK

**The heavy-tailed phenomenon.** Heavy tails in machine learning have been observed and studied in various forms. The most prominent empirical results are from a series of works by Martin & Mahoney (2020; 2021a;b); Martin et al. (2021), which propose a heavy-tailed self regularization (HT-SR) theory of deep NNs. In particular, Martin & Mahoney (2021a) proposed a '5 + 1' phase model corresponding to the ESD evolution, and aims to model the HT-SR effect during training. Their observations from extensive empirical analysis showcased a correlation between the power law fits of weight matrix ESDs and generalization. Although it is still unclear if HT ESDs are necessary for generalization, Martin et al. (2021); Yang et al. (2023) have shown that the shape metrics of the ESD can be effectively leveraged to identify well-trained NNs (and can be extended to large-scale models such as LLMs). From a theoretical perspective, previous efforts have primarily focused on stochastic optimization settings and proposed generalization bounds based on the tail indices of the stochastic noise (Simsekli et al., 2019; 2020b;a; Gurbuzbalaban et al., 2021; Hodgkinson & Mahoney, 2021; Hodgkinson et al., 2022; Raj et al., 2023; Nguyen et al., 2019; Barsbey et al., 2021). In particular, the earlier work by Simsekli et al. (2020a) employed continuous-time approximation of `SGD` via feller-processes and studied the role of the Hausdorff Dimension on generalization. More recently, Hodgkinson et al. (2022) extended this analysis to discrete-time settings. In concurrent works, Gurbuzbalaban et al. (2021) studied the heavy-tailed stationary distributions of discrete Markov processes as an approximation to SGD iterates in the infinite data regime, while Hodgkinson & Mahoney (2021) analyzed the role of multiplicative noise in such settings. While these efforts have emphasized the role of the learning rate/batch size ratio in determining the tail index of the iterates, a formal study on the role of these hyper-parameters is still lacking. More importantly, none of these efforts have focused on the evolution of the ESD itself. Although the limiting HT distribution of the weight values can lead to HT ESDs (Arous & Guionnet, 2008), the insights cannot be directly used for a fine-grained analysis of the '5 + 1' phase model. Our work aims to bridge these gaps and analyzes the fine-grained ESD evolution from a Marchenko-Pastur (MP) fit (i.e. random initialization) → "Bulk+Spike" → "Bulk-Decay" → HT distributions and the correlations with generalization.

**Feature learning and large learning rates.** Learning *single-index* models using two-layer NNs under the *Teacher-Student* setup has provided rich insights into the sample complexity (Damian et al., 2023; Mousavi-Hosseini et al., 2023a; Zweig et al., 2023; Damian et al., 2024; 2022; Abbe et al., 2023) and training dynamics (Bietti et al., 2022; Wang et al., 2023; Cui et al., 2024; Moniri et al., 2023) of NNs. In particular, the study of *feature learning* in such two-layer NNs focuses on the factors (such as optimizers (Abbe et al., 2023), loss landscapes (Damian et al., 2023), representations (Nichani et al., 2023) and learning rates (Dandi et al., 2023)) that facilitate sample efficient learning beyond the kernel regime (Louart et al., 2018; Gerace et al., 2020; Mei & Montanari, 2022; Hu & Lu, 2022; Goldt et al., 2022; Liu et al., 2021). Recently, Ba et al. (2022) analyzed the first step of `GD` update in the high dimensional setting and formalized the scale of $\eta$ required to go beyond the random feature regime. In particular, such a *large* $\eta$ is necessary for a two-layer NN to learn the hidden direction of a single-index model after one step of `GD` (see also (Dandi et al., 2023)). This first *large* update was shown to result in an outlier singular value (i.e. a 'spike') in the ESD of the hidden layer weight matrix. Our work presents the first result for such *large* $\eta$ in the case of `FB-Adam` and can be of wider interest in the feature learning context. Beyond the one-step analysis, Dandi et al. (2024) analyzed the role of two-pass `GD` in learning single-index models with large information

exponents (Arous et al., 2021). However, the effects of multiple passes over the data on the ESD and generalization are yet to be fully understood. Our multi-step analysis empirically highlights the importance of such feature learning after the first step for the emergence of HT ESD. Thus, showcasing the unexplored connections between these two areas of research.

# 3 PRELIMINARIES AND SETUP

**Notation.** For $n \in \mathbb{N}$, we denote $[n] = \{1, \cdots, n\}$. We use $O(\cdot)$ to denote the standard big-O notation and the subscript $O_d(\cdot)$ to denote the asymptotic limit of $d \to \infty$. Formally, for two sequences of real numbers $x_d$ and $y_d$, $x_d = O_d(y_d)$ represents $\lim_{d\to\infty} |x_d| \leq C_1|y_d|$ for some constant $C_1$. Similarly, $x_d = O_{d,\mathbb{P}}(y_d)$ denotes that the asymptotic inequality almost surely holds under a probability measure $\mathbb{P}$. The definitions can be extended to the standard $\Omega(\cdot), \Theta(\cdot)$ or $\asymp$ notations analogously (Graham & Knuth, 1989). For two sequences of real numbers $x_d$ and $y_d$, $x_d \asymp y_d$ represents $|y_d|C_2 \leq |x_d| \leq C_1|y_d|$, for constants $C_1, C_2 > 0$ (Wang et al., 2021; Moniri et al., 2023). For a real matrix $\boldsymbol{B} = (B_{ij})_{n\times m} \in \mathbb{R}^{n\times m}$, $\boldsymbol{B}^{\circ p}$ represents an element-wise $p$-power transformation such that $\boldsymbol{B}^{\circ p} = (B_{ij}^p)_{n\times m}$. $\odot$ is the matrix Hadamard product, $\text{sign}(.)$ denotes the element-wise sign function. $\|\cdot\|_2$ denotes the $\ell_2$ norm for vectors and the operator norm for matrices. $\|\cdot\|_F$ denotes the Frobenius norm. $\boldsymbol{0}_{h\times d}, \boldsymbol{1}_{h\times d} \in \mathbb{R}^{h\times d}$ represent the all-zero and all-ones matrices.

**Dataset.** We sample $n$ data points $\{\boldsymbol{x}_1, \cdots, \boldsymbol{x}_n\}$ from the isotropic Gaussian $\boldsymbol{x}_i \sim \mathcal{N}(\boldsymbol{0}_d, \boldsymbol{I}_d), \forall i \in [n]$ as our input data. For a given $\boldsymbol{x}_i \in \mathbb{R}^d$, we use a single-index *teacher* model $F^* : \mathbb{R}^d \to \mathbb{R}$ to generate the corresponding scalar label $y_i \in \mathbb{R}$ as follows:

$$y_i = F^*(\boldsymbol{x}_i) + \xi_i = \sigma_*(\boldsymbol{\beta}^{*\top}\boldsymbol{x}_i) + \xi_i. \tag{1}$$

Here $\boldsymbol{\beta}^* \in \mathbb{S}^{d-1}$ (the $d-1$-dimensional sphere in $\mathbb{R}^d$) is the *target direction*, $\sigma_* : \mathbb{R} \to \mathbb{R}$ is the *target non-linear link function*, and $\xi_i \sim \mathcal{N}(0, \rho_e^2)$ is the independent additive label noise. We represent $\boldsymbol{X} \in \mathbb{R}^{n\times d}, \boldsymbol{y} \in \mathbb{R}^n$ as the input matrix and the label vector, respectively.

**Learning.** We consider a two-layer fully-connected NN with activation $\sigma : \mathbb{R} \to \mathbb{R}$ as our *student* model $f(\cdot) : \mathbb{R}^d \to \mathbb{R}$. For an input $\boldsymbol{x}_i \in \mathbb{R}^d$, its prediction is formulated as:

$$f(\boldsymbol{x}_i) = \frac{1}{\sqrt{h}}\boldsymbol{a}^\top \sigma\left(\frac{1}{\sqrt{d}}\boldsymbol{W}\boldsymbol{x}_i\right). \tag{2}$$

Here $\boldsymbol{W} \in \mathbb{R}^{h\times d}, \boldsymbol{a} \in \mathbb{R}^h$ are the first and second layer weights, respectively, with entries sampled i.i.d as follows $[\boldsymbol{W}_0]_{i,j} \sim \mathcal{N}(0, 1), [a]_i \sim \mathcal{N}(0, 1), \forall i \in [h], j \in [d]$.

## 3.1 TRAINING PROCEDURE

We employ the following *Two-stage training* procedure (Ba et al., 2022; Moniri et al., 2023; Cui et al., 2024; Dandi et al., 2023; Wang et al., 2023) on the student network. In the first stage, we fix the last layer weights $\boldsymbol{a} \in \mathbb{R}^h$ and apply optimizer update(s) (GD/FB-Adam) only for the first layer $\boldsymbol{W}$. In the second stage, we perform ridge regression on the last layer using a hold-out dataset of the same size to calculate the ideal value of $\boldsymbol{a}$ (Ba et al., 2022).

**Optimizer updates for the first layer.** In this phase, we fix the last layer weights $\boldsymbol{a}$ to its value at initialization and perform GD/FB-Adam update(s) on $\boldsymbol{W}$ to minimize the mean-squared error $R(f, \boldsymbol{X}, \boldsymbol{y}) = \frac{1}{2n}\sum_{i=1}^n (y_i - f(\boldsymbol{x}_i))^2$. The update to $\boldsymbol{W}$ using GD is given by:

$$\boldsymbol{W}_{t+1} = \boldsymbol{W}_t - \eta\boldsymbol{G}_t, \tag{3}$$

where $\boldsymbol{W}_t$ denotes the weights $\boldsymbol{W}$ at step $t$ and $\boldsymbol{G}_t = \nabla_{\boldsymbol{W}_t}R(f, \boldsymbol{X}, \boldsymbol{y})$ represents the full-batch gradient. Next, to formulate the updates using FB-Adam, let:

$$\widetilde{\boldsymbol{M}}_{t+1} = \beta_1\widetilde{\boldsymbol{M}}_t + (1-\beta_1)\boldsymbol{G}_t, \qquad \widetilde{\boldsymbol{V}}_{t+1} = \beta_2\widetilde{\boldsymbol{V}}_t + (1-\beta_2)\boldsymbol{G}_t^{\circ 2}. \tag{4}$$

Here $\widetilde{\boldsymbol{M}}_t, \widetilde{\boldsymbol{V}}_t \in \mathbb{R}^{h\times d}$ represent the first and second order moving averages of the gradient respectively, with base values $\widetilde{\boldsymbol{M}}_0 = \boldsymbol{0}_{h\times d}, \widetilde{\boldsymbol{V}}_0 = \boldsymbol{0}_{h\times d}$ (Kingma & Ba, 2014). $(\beta_1, \beta_2) \in \mathbb{R}$ are the decay

factors. Considering $\widetilde{\boldsymbol{G}}_t = (\widetilde{\boldsymbol{V}}_{t+1}^{\circ 1/2} + \epsilon \mathbf{1}_{h \times d})^{\circ -1} \odot \widetilde{\boldsymbol{M}}_{t+1}$, we formulate the `FB-Adam` update[1]:

$$\boldsymbol{W}_{t+1} = \boldsymbol{W}_t - \eta \widetilde{\boldsymbol{G}}_t. \tag{5}$$

For the remainder of this paper, we use the overloaded term *'optimizer update'* to represent either `FB-Adam` or `GD` update. Specific choices of the optimizer will be mentioned explicitly.

**Ridge-regression on final layer.** Similar to the setup of Ba et al. (2022); Moniri et al. (2023); Wang et al. (2023); Ba et al. (2023), we consider a hold-out training dataset $\overline{\boldsymbol{X}} \in \mathbb{R}^{n \times d}, \overline{\boldsymbol{y}} \in \mathbb{R}^n$ sampled in the same fashion as $\boldsymbol{X}, \boldsymbol{y}$ to learn the last layer weights. Formally, after $t$ optimizer updates to the first layer to obtain $\boldsymbol{W}_t$, we calculate the post-activation features $\overline{\boldsymbol{Z}}_t = \frac{1}{\sqrt{h}} \sigma \left( \frac{1}{\sqrt{d}} \boldsymbol{W}_t \overline{\boldsymbol{X}}^\top \right)$ and solve the following ridge-regression problem:

$$\hat{\boldsymbol{a}} = \arg \min_{\boldsymbol{a} \in \mathbb{R}^h} \frac{1}{n} \left\| \overline{\boldsymbol{y}} - \overline{\boldsymbol{Z}}_t^\top \boldsymbol{a} \right\|_2^2 + \frac{\lambda}{h} \|\boldsymbol{a}\|_2^2. \tag{6}$$

Here $\lambda > 0$ is the regularization constant. The solution $\hat{\boldsymbol{a}}$ is now used as the last-layer weight vector for our student network $f(\cdot)$ and we consider the resulting regression loss as our *training loss*. Formally, this setup allows us to measure the impact of updates to $\boldsymbol{W}$ after $t$ steps on the hold-out dataset's regression loss. Finally, for a test sample $\boldsymbol{x} \in \mathbb{R}^d$, the student network prediction is given as: $\hat{y} = \frac{1}{\sqrt{h}} \hat{\boldsymbol{a}} \sigma \left( \frac{1}{\sqrt{d}} \boldsymbol{W}_t \boldsymbol{x} \right)$. These predictions on the test data are used for computing the *test loss* using the mean squared error.

### 3.2 ALIGNMENT AND HT METRICS

**Alignment between $\boldsymbol{W}, \boldsymbol{\beta}^*$.** To quantify the "extent" of feature learning in our student network during training, we measure the alignment between the first principal component of $\boldsymbol{W}$ (denoted as $\boldsymbol{u}_1$) and the target direction $\boldsymbol{\beta}^*$ (Ba et al., 2022; Wang et al., 2023) as: $\texttt{sim}(\boldsymbol{W}, \boldsymbol{\beta}^*) = |\boldsymbol{u}_1^\top \boldsymbol{\beta}^*|$.

**Kernel Target Alignment (`KTA`).** In addition to analyzing $\boldsymbol{W}$, we also consider the alignment between the Conjugate Kernel (*CK*) (Wang et al., 2023; Lee et al., 2018; Matthews et al., 2018; Fan & Wang, 2020) based on hidden layer activations and the target outputs. Formally, consider the hidden layer activations of the holdout data as $\overline{\boldsymbol{Z}} = \frac{1}{\sqrt{h}} \sigma \left( \frac{1}{\sqrt{d}} \boldsymbol{W} \overline{\boldsymbol{X}}^\top \right)$ and define *CK* as: $\boldsymbol{K} = \overline{\boldsymbol{Z}}^\top \overline{\boldsymbol{Z}} \in \mathbb{R}^{n \times n}$. The `KTA` (Cristianini et al., 2001) between $\boldsymbol{K}$ and $\boldsymbol{yy}^\top \in \mathbb{R}^{n \times n}$ is given as:

$$\texttt{KTA} = \frac{\langle \boldsymbol{K}, \boldsymbol{yy}^\top \rangle}{\|\boldsymbol{K}\|_F \|\boldsymbol{yy}^\top\|_F}, \quad \langle \boldsymbol{K}, \boldsymbol{yy}^\top \rangle = \sum_{i,j}^n K_{i,j} (yy^\top)_{i,j}. \tag{7}$$

**Power-law fits (`PL_Alpha_Hill`, `PL_Alpha_KS`)** To quantify the heaviness of the tails in the ESD, we measure `PL_Alpha_Hill` (Zhou et al., 2023), and `PL_Alpha_KS`, which refer to the power-law exponents that are fit to the ESD of $\boldsymbol{W}_t^\top \boldsymbol{W}_t$ using the Hill estimator (Hill, 1975) and based on the Kolmogorov–Smirnoff statistic respectively (Martin et al., 2021; Clauset et al., 2009).

## 4 BULK+SPIKE PHENOMENON AFTER ONE OPTIMIZER STEP

In this section, we present empirical and theoretical results for the scale of $\eta$ for `FB-Adam` that results in the "Bulk+Spike" ESD and facilitates feature learning in the student network after the first update. The discussion on the first optimizer update and $\eta$ is essential, as we show in Section 5 that the Bulk+Spike ESD after the first update facilitates the emergence of HT ESDs.

### 4.1 THE BULK+SPIKE ESD BY SCALING $\eta$

**Setup.** We consider the two-layer NN $f(\cdot)$ of width $h = 1500, \sigma = \texttt{tanh}$ and train it on a dataset of size $n = 2000$, with input dimension $d = 1000$. We choose $\sigma_* = \texttt{softplus}$ as the target link function and set the label noise to $\rho_e = 0.3$, with $\lambda = 0.01$. The test data consists of 200 samples.

---

[1] We choose the subscript $t$ in $\widetilde{\boldsymbol{G}}_t$ for notational consistency between `FB-Adam` and `GD` updates.

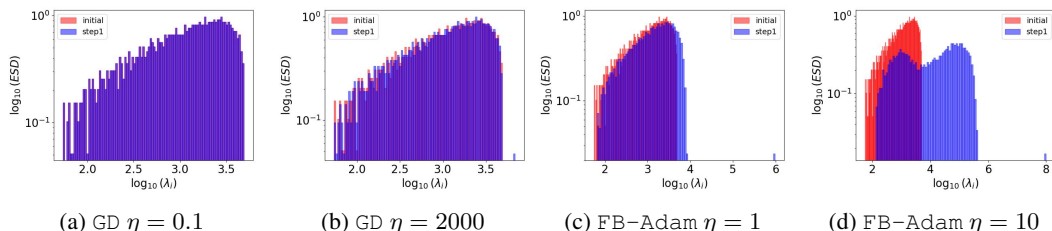

(a) GD $\eta = 0.1$     (b) GD $\eta = 2000$     (c) FB-Adam $\eta = 1$     (d) FB-Adam $\eta = 10$

Figure 2: ESD of $\boldsymbol{W}_1^\top \boldsymbol{W}_1$ for GD/FB-Adam with varying $\eta$, and $n = 2000, d = 1000, h = 1500, \sigma_* = \texttt{softplus}, \sigma = \texttt{tanh}, \rho_e = 0.3, \lambda = 0.01$.

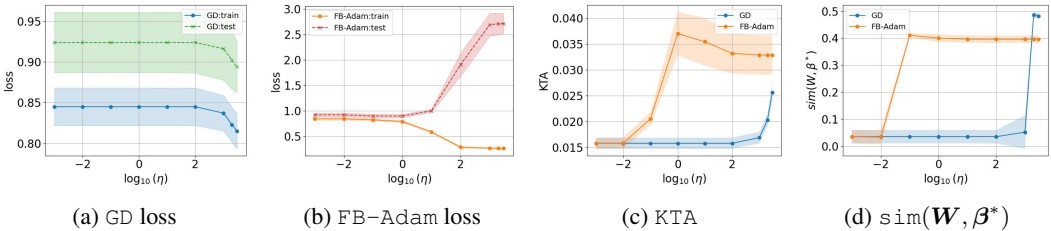

(a) GD loss     (b) FB-Adam loss     (c) KTA     (d) $\text{sim}(\boldsymbol{W}, \boldsymbol{\beta}^*)$

Figure 3: Losses, KTA, $\text{sim}(\boldsymbol{W}, \boldsymbol{\beta}^*)$ for $f(\cdot)$ trained with one-step of GD, FB-Adam. Here $n = 2000, d = 1000, h = 1500, \sigma_* = \texttt{softplus}, \sigma = \texttt{tanh}, \rho_e = 0.3, \lambda = 0.01$.

**FB-Adam needs a much smaller $\eta$ than GD to exhibit a Bulk+Spike ESD.** By varying $\eta$, we train the student network $f(\cdot)$ for one step using GD and FB-Adam. We can observe from Figure 2a that after one GD update with $\eta = 0.1$, the ESD of $\boldsymbol{W}_1^\top \boldsymbol{W}_1$ remains largely unchanged from that of $\boldsymbol{W}_0^\top \boldsymbol{W}_0$ (i.e ESD at random initialization). On the other hand, Figure 2b illustrates that for $\eta = 2000$, the ESD of $\boldsymbol{W}_1^\top \boldsymbol{W}_1$ exhibits a spike. However, FB-Adam exhibits a spike in the ESD of $\boldsymbol{W}_1^\top \boldsymbol{W}_1$ after the first step even with $\eta = 1$ (see Figure 2c). Finally, for $\eta = 10$, the ESD tends towards a seemingly bimodal distribution (see Figure 2d).

**Impact on losses, KTA, and $\text{sim}(\boldsymbol{W}, \boldsymbol{\beta}^*)$.** As the choice of optimizer affects the scale of $\eta$ leading to a Bulk+Spike ESD of $\boldsymbol{W}_1^\top \boldsymbol{W}_1$, we vary $\eta$ across $\{0.001, 0.01, 0.1, 1, 10, 100, 1000, 2000, 3000\}$ and plot the means and standard deviations of losses, KTA and $\text{sim}(\boldsymbol{W}, \boldsymbol{\beta}^*)$ across 5 runs in Figure 3. In the case of GD, observe that $\eta = 2000$ is the threshold for the: reduction of train and test losses (Figure 3a), an increase in KTA (Figure 3c), and an increase in $\text{sim}(\boldsymbol{W}, \boldsymbol{\beta}^*)$ (Figure 3d). Thus, implying that the occurrence of a spike leads to better generalization after one step (as also verified by Ba et al. (2022)). In the case of FB-Adam, $\eta < 0.01$ does not improve generalization and we do not see an increase in KTA/$\text{sim}(\boldsymbol{W}, \boldsymbol{\beta}^*)$. For $0.01 \leq \eta \leq 1$, generalization improves, and KTA/$\text{sim}(\boldsymbol{W}, \boldsymbol{\beta}^*)$ values increase. For $\eta > 1$, generalization degrades and there is a slight reversal in the trend for KTA. Note that due to the two-phase training strategy, the large magnitude of the first step update with $\eta > 1$ can lead to lower training loss (since the last layer is computed based on a closed-form solution), but lead to higher test loss. These observations hint at a sweet spot for $\eta$ beyond which FB-Adam leads to poor test performance (see also Appendix E.1). Towards understanding these observations, we focus on the following pressing question: *why does $\eta = 0.1$ suffice for one-step of FB-Adam to exhibit a spike in the ESD, whereas one-step GD requires $\eta = 2000$?* Specifically, how large should $\eta$ be for FB-Adam to exhibit a spike in the ESD?

### 4.2 THEORETICAL RESULTS FOR SCALING $\eta$ WITH FB-ADAM

To theoretically answer the above questions, we start by formulating the first step gradient $\boldsymbol{G}_0$ as:

$$\boldsymbol{G}_0 = \frac{1}{n\sqrt{d}} \left[ \frac{1}{\sqrt{h}} \left( \boldsymbol{a}\boldsymbol{y}^\top - \frac{1}{\sqrt{h}} \boldsymbol{a}\boldsymbol{a}^\top \sigma \left( \frac{1}{\sqrt{d}} \boldsymbol{W}_0 \boldsymbol{X}^\top \right) \right) \odot \sigma' \left( \frac{1}{\sqrt{d}} \boldsymbol{X} \boldsymbol{W}_0^\top \right) \right] \boldsymbol{X}. \quad (8)$$

Here $\sigma'(\cdot) : \mathbb{R} \to \mathbb{R}$ is the derivative of the activation function $\sigma$ acting element-wise on $\boldsymbol{X}\boldsymbol{W}_0^\top / \sqrt{d}$. Based on equation equation 4, let $\widetilde{\boldsymbol{P}}_1 = \widetilde{\boldsymbol{V}}_1^{\circ 1/2} + \epsilon \mathbf{1}_{h \times d}$. Considering the FB-Adam epsilon hyperparameter $\epsilon \approx 0$ (Kunstner et al., 2022), the update can be given as:

$$\widetilde{\boldsymbol{G}}_0 = \widetilde{\boldsymbol{P}}_1^{\circ -1} \odot \widetilde{\boldsymbol{M}}_1 = \frac{1 - \beta_1}{\sqrt{1 - \beta_2}} \text{sign}(\boldsymbol{G}_0). \quad (9)$$

**Theorem 4.1** *Given the two-stage training procedure with large $n, d$, such that $n \asymp d$, and large (fixed) $h$, assume the teacher $F^*$ is $\lambda_\sigma$-Lipschitz with $\|F^*\|_{L^2} = \Theta_d(1)$, and a normalized 'student' activation $\sigma$, which has $\lambda_\sigma$-bounded first three derivatives almost surely and satisfies $\mathbb{E}[\sigma(z)] = 0, \mathbb{E}[z\sigma(z)] \neq 0$, for $z \sim \mathcal{N}(0,1)$; then the matrix norm bounds for the one-step* `FB-Adam` *update can be given as:*

$$\left\|\widetilde{\boldsymbol{G}}_0\right\|_2 = \Theta_{d,\mathbb{P}}(\sqrt{hd}), \quad \left\|\widetilde{\boldsymbol{G}}_0\right\|_F = \Theta_{d,\mathbb{P}}(\sqrt{hd}). \quad (10)$$

Appendix B presents the proof. The sketch of the proof is as follows: First, we show that $\boldsymbol{G}_0$ does not contain values that are exactly equal to 0 almost surely. This allows us to obtain $\left\|\widetilde{\boldsymbol{G}}_0\right\|_F = \Theta_{d,\mathbb{P}}(\sqrt{hd})$. Next, we leverage a rank 1 approximation $\boldsymbol{A}$ of $\boldsymbol{G}_0$ (in the operator norm (Ba et al., 2022)) to show that $\text{sign}(\boldsymbol{A}) = \text{sign}(\boldsymbol{G}_0)$. Finally, we obtain the lower-bound of $\left\|\widetilde{\boldsymbol{G}}_0\right\|_2 = \Omega_{d,\mathbb{P}}(\sqrt{hd})$ to prove the theorem [2].

**Corollary 4.2** *Under the assumptions of Theorem 4.1, we have the following learning rate scales*

$$\eta = \Theta(1) \implies \|\boldsymbol{W}_1 - \boldsymbol{W}_0\|_F \asymp \|\boldsymbol{W}_0\|_F$$
$$\eta = \Theta\left(1/\sqrt{h}\right) \implies \|\boldsymbol{W}_1 - \boldsymbol{W}_0\|_2 \asymp \|\boldsymbol{W}_0\|_2, \quad (11)$$

*where $\|\boldsymbol{W}_0\|_2 = \Theta_{d,\mathbb{P}}(\sqrt{d}), \|\boldsymbol{W}_0\|_F = \Theta_{d,\mathbb{P}}(\sqrt{hd})$.*

Theorem 4.1 shows that the spectral and Frobenius norms of $\widetilde{\boldsymbol{G}}_0$ scale similarly and that the top singular value contributes the most to the Frobenius norm as $d, h$ increase. As a consequence, Corollary 4.2 indicates that $\eta = \Theta(1)$ (which is independent of $h, d$) is sufficient for $\eta\widetilde{\boldsymbol{G}}_0$ to result in a 'spike' in the ESD of $\boldsymbol{W}_1^\top \boldsymbol{W}_1$. This explains our empirical observations above where even $\eta = 1$ was sufficient for the ESD to transition into a Bulk+Spike shape.

**Remark.** We note that a similar result for $\eta$ in the case of `GD` was previously established by Ba et al. (2022). However, they employ a mean-field initialization of the two-layer NN $f(\cdot)$ and differs from our NTK-based initialization (Wang et al., 2023). Nonetheless, we show that our results can be extended to such a setup as well and the adjusted $\eta$ can indeed explain the Bulk+Spike ESD after one step update (Appendix C presents a comprehensive discussion).

## 5 HT PHENOMENON AFTER MULTIPLE OPTIMIZER STEPS

In this section, we empirically analyze the ESD evolution from the "Bulk+Spike" shape into an HT distribution. This is followed by a correlation analysis with generalization of the two-layer NN and finally presents a singular vector perspective on the HT ESD emergence.

### 5.1 TRANSITIONING FROM BULK+SPIKE TO HT ESD

**Spike after one step facilitates the emergence of HT ESD.** We begin by employing the same setup as Section 4 and show in Table 1 that the spike in the ESD after one step facilitates the emergence of HT ESD. We observe that after just 10 `FB-Adam` steps with $\eta = 1$, the HT ESD emerges. However, $\eta = 0.1$ requires $t = 10000$. This indicates that the relative position of the spike from the bulk (which is captured by the spectral gap in this setup) plays a key role in determining the step complexity of the HT ESD emergence [3]. Finally, for $\eta = 0.01$, which does not exhibit an HT ESD even after $t = 10000$, we observed that a significantly long amount of training for $t = 10^6$ steps can lead to HT. Note that $\eta = 0.01$ tends to fall within the $(\Theta(1/\sqrt{h}), \Theta(1))$ range, considering the scaling constants for $\Theta$. This leads to a scenario where a spike does not emerge after one step but the magnitude of gradient updates over $t = 10^6$ steps can lead to HT ESD (see Table 5). Given this observation, extremely small $\eta$ might not be able to exhibit such changes to the ESD even after $t > 10^6$ steps (see Corollary 5.3 in Wang et al. (2023) for a related formal result). We extend the analysis to a wider range of $\eta$ in Appendix D (see Table 4).

---

[2]The normalized activation is not a practical limitation and is solely required for the proof (Ba et al., 2022).

[3]Theoretical analysis of the spectral gap and its role on the step complexity for HT ESD emergence is out of scope of this paper and can be a valuable direction for future research.

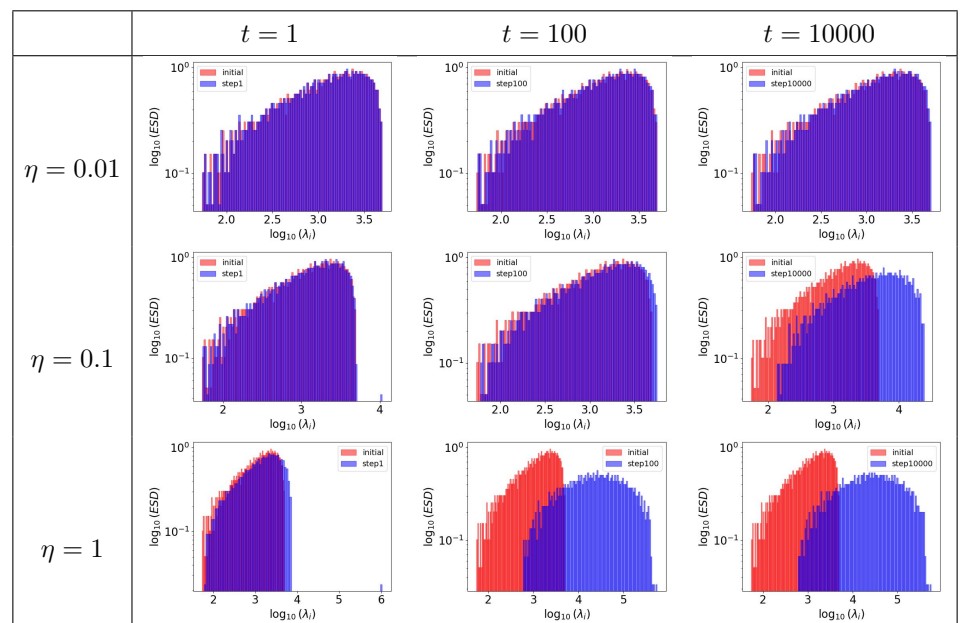

Table 1: Evolution of ESD over steps $t \in \{1, 100, 10000\}$ with $\eta \in \{0.01, 0.1, 1\}$ for `FB-Adam`.

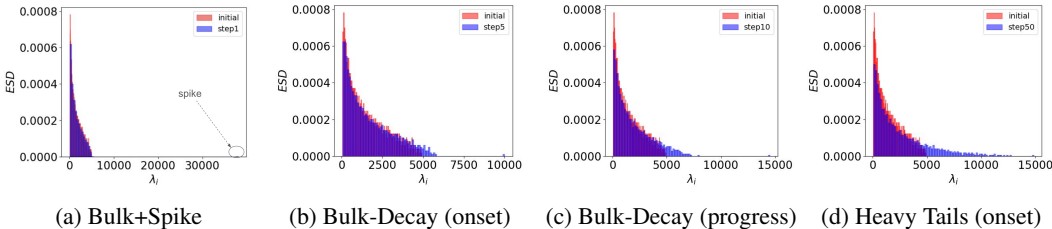

(a) Bulk+Spike     (b) Bulk-Decay (onset)     (c) Bulk-Decay (progress)     (d) Heavy Tails (onset)

Figure 4: Evolution of ESD (linear-linear scale) from a Marchenko-Pastur fit (at initialization $t = 0$: red bulk) to (a) Bulk+Spike ($t = 1$), (b) the onset of Bulk-Decay ($t = 5$), (c) Continued decay of the bulk ($t = 10$), (d) onset of heavy tails ($t = 50$), with `FB-Adam` optimizer and $\eta = 0.2$.

**Bulk-Decay as an intermediate stage.** The '5+1' model of the ESD evolution in deep NNs (Martin & Mahoney, 2021a) is characterized by the following phases: (1) Random-like (2) Bleeding-out (3) Bulk+Spikes (4) Bulk-Decay (5) Heavy-Tailed (HT) and the final (6) Rank-Collapse[4]. By considering `FB-Adam` with $\eta = 0.2$ and the same setup as Section 4, we illustrate in Figure 4 that the spike emerges after one-step (Figure 4a) and gradually decays the Bulk (Figure 4b, Figure 4c) toward a HT distribution (Figure 4d). Since practical settings employ deep NNs and train with stochastic gradient noise on complex datasets, the ESD evolution is much more nuanced. Especially the Bleeding-out phase and the presence of multiple spikes before the onset of Bulk-Decay. Nonetheless, we extended our results from the student-teacher setup to more practical settings, such as VGG models on MNIST, and validated our HT ESD findings in those settings as well (Appendix E.6). Overall, we emphasize that our setup with two-layer NNs and single-index models opens up the possibility to approximate such dynamics while being subject to theoretical treatment.

## 5.2 IMPACT ON GENERALIZATION

**Setup.** We consider the two-layer student NN $f(\cdot)$ of width $h = 1500, \sigma = $ `tanh` and train it on a dataset of size $n = 8000$ for 10 steps using `GD`/`FB-Adam`. We choose a sample dimension $d = 1000, \sigma_* = $ `softplus` and $\rho_e = 0.3$. The test dataset has 200 samples.

---

[4]The rank-collapse is an extreme case scenario which is not considered in our analysis.

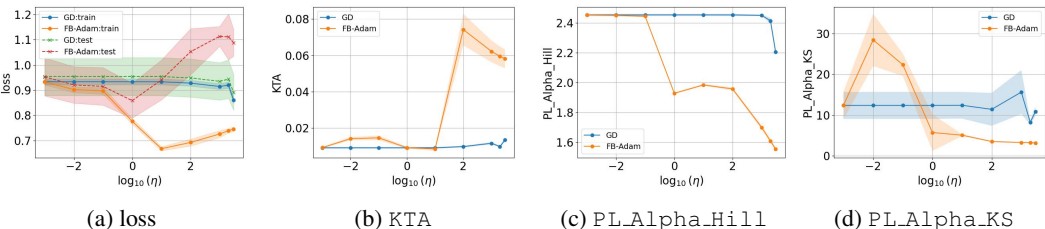

(a) loss      (b) KTA      (c) PL_Alpha_Hill      (d) PL_Alpha_KS

Figure 5: Losses, KTA, PL_Alpha_Hill, PL_Alpha_KS after 10 steps of GD, FB-Adam, with $n = 8000, d = 1000, h = 1500, \sigma_* = \texttt{softplus}, \sigma = \texttt{tanh}, \rho_e = 0.3, \lambda = 0.01$.

**Correlations between ESD and losses.** We plot the means and standard deviations across 5 runs for the train/test losses, KTA, PL_Alpha_Hill and PL_Alpha_KS (see Section 3.2) of $W_{10}^\top W_{10}$ after 10 GD/FB-Adam updates by varying $\eta$ across $\{0.001, 0.01, 0.1, 1, 10, 100, 1000, 2000, 3000\}$ in Figure 5. A lower value of PL_Alpha_Hill / PL_Alpha_KS indicates a heavier-tailed spectrum[5]. Observe that for baseline GD experiments with $\eta \geq 1000$, the reduction in train and test losses are correlated with an increase in KTA and a decrease in PL_Alpha_Hill and PL_Alpha_KS. Additional experiments are presented in Appendix E.2.

In the case of FB-Adam, a much clearer correlation between the training loss, KTA, PL_Alpha_Hill and PL_Alpha_KS can be observed. Especially, there seems to be a region of benign learning rates ($0.01 \leq \eta \leq 1$) for which, the PL_Alpha_Hill estimates lie in the range of $(2, 2.5)$ and a decrease in the estimate (resulting in a 'heavier' tailed ESD) improves generalization. For $1 \leq \eta \leq 100$, although we observe similar values of PL_Alpha_Hill, the ESDs of $W_{10}^\top W_{10}$ differ in the scale of the singular values, and the spike seems to have a large influence on the estimation of PL_Alpha_Hill (see Figure 23 in Appendix E). However, the PL_Alpha_KS captures the monotonically decreasing trend for this range of $\eta$. Finally, for extremely large $\eta > 100$, we observe much smaller estimates of PL_Alpha_Hill ($< 1.8$) but these extremely heavier tails do not correlate with better generalization. We also note that these benign $\eta$ ranges vary based on the choice of the activation functions. In particular, when $\sigma = \sigma_* = \texttt{tanh}$, we observed that the range can be reduced by an order of magnitude (Appendix E.5). Overall, these observations support the conclusions of Martin & Mahoney (2021a); Martin et al. (2021) which state that well-trained deep NNs do not exhibit extreme HT ESDs but rather whose power-law estimates lie within a suitable range (see also Appendix E.7). For instance, a range of $(2, 4)$ for the truncated power-law fit estimates.

**Remark.** Since practical training approaches employ techniques such as weight normalization (WN) and learning rate schedules, we present a preliminary analysis of their role in the ESD evolution and generalization in Appendix E.4. We employ a WN technique (Huang et al., 2023) after each update: $W_{t+1} = \frac{\sqrt{hd}W'_{t+1}}{\|W'_{t+1}\|_F}, W'_{t+1} = W_t + M_t$, to ensure that $\|W_{t+1}\|_F$ is always $\sqrt{hd}$, before the forward pass. In summary, we observed that employing WN leads to relatively heavier-tailed spectra (i.e. lower PL_Alpha_Hill) while exhibiting similar correlations with generalization as discussed above. On the other hand, by employing schedulers such as torch.optim.StepLR, we showcase a fine-grained manipulation of the ESD evolution depending on the decay rate $\gamma$.

## 5.3 SINGULAR VECTOR ALIGNMENTS OF WEIGHTS AND OPTIMIZER UPDATES

In this section, we present a singular-vector perspective on the emergence of HT ESD after multiple update steps. By considering an 'update' matrix $M_t \in \mathbb{R}^{h \times d}$, we formulate the weight updates as:

$$W_{t+1} = W_t + M_t, \tag{12}$$

where $M_t$ is the optimizer update matrix based on GD ($M_t = -\eta G_t$) or FB-Adam ($M_t = -\eta \widetilde{G}_t$). By abstracting $W_{t+1}$ as the *'observation'*, $W_t$ as the *'noise'* and $M_t$ as the *'signal'* rectangular matrices, we leverage methods from the rich literature on signal recovery in spiked matrix models (Shabalin & Nobel, 2013; Landau et al., 2023; El Alaoui & Jordan, 2018; Gavish & Donoho, 2017; Troiani et al., 2022) to analyze the role of singular vectors of $M_t$ in transforming the ESD of $W_t^\top W_t$

---

[5]The effects of the power-law exponent estimation approaches are discussed in Appendix F.

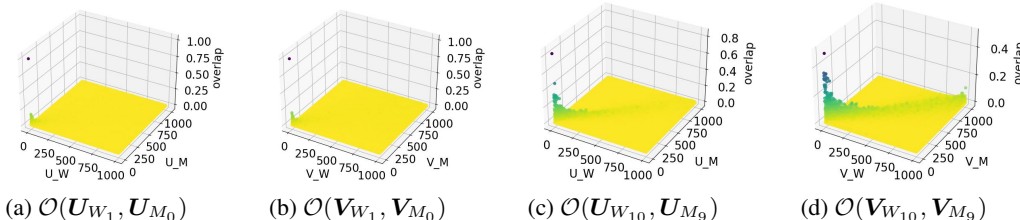

(a) $\mathcal{O}(\boldsymbol{U}_{W_1}, \boldsymbol{U}_{M_0})$  (b) $\mathcal{O}(\boldsymbol{V}_{W_1}, \boldsymbol{V}_{M_0})$  (c) $\mathcal{O}(\boldsymbol{U}_{W_{10}}, \boldsymbol{U}_{M_9})$  (d) $\mathcal{O}(\boldsymbol{V}_{W_{10}}, \boldsymbol{V}_{M_9})$

Figure 6: Overlaps after one step (a), (b) and 10 steps (c), (d) of `FB-Adam`($\eta = 1$).

gradually into a HT distribution. Let $b = \min(h, d)$, we consider the SVD of $\boldsymbol{W}_{t+1}, \boldsymbol{W}_t, \boldsymbol{M}_t$ as:

$$\boldsymbol{W}_{t+1} = \boldsymbol{U}_{W_{t+1}} \boldsymbol{S}_{W_{t+1}} \boldsymbol{V}_{W_{t+1}}^\top, \quad \boldsymbol{W}_t = \boldsymbol{U}_{W_t} \boldsymbol{S}_{W_t} \boldsymbol{V}_{W_t}^\top, \quad \boldsymbol{M}_t = \boldsymbol{U}_{M_t} \boldsymbol{S}_{M_t} \boldsymbol{V}_{M_t}^\top, \quad (13)$$

where $\boldsymbol{U}_{W_{t+1}}, \boldsymbol{U}_{W_t}, \boldsymbol{U}_{M_t} \in \mathbb{R}^{h \times b}$, $\boldsymbol{S}_{W_{t+1}}, \boldsymbol{S}_{W_t}, \boldsymbol{S}_{M_t} \in \mathbb{R}^{b \times b}$ and $\boldsymbol{V}_{W_{t+1}}, \boldsymbol{V}_{W_t}, \boldsymbol{V}_{M_t} \in \mathbb{R}^{b \times d}$.

**Definition 5.1** *(Landau et al., 2023). The 'overlaps' between two singular vector matrices $\boldsymbol{J}, \boldsymbol{Q} \in \mathbb{R}^{a \times b}$ is defined as: $\mathcal{O}(\boldsymbol{J}, \boldsymbol{Q}) = (\boldsymbol{J}^\top \boldsymbol{Q})^{\circ 2} \in \mathbb{R}^{b \times b}$*

**Overlaps during training.** From the finite rank spiked matrix model, we know that the singular values $\boldsymbol{S}_{W_1}$ are non-linear transformations (also termed as 'inflations' (Landau et al., 2023)) of $\boldsymbol{S}_{M_0}$, and $\boldsymbol{U}_{W_1}, \boldsymbol{V}_{W_1}$ are rotated variants of $\boldsymbol{U}_{M_0}, \boldsymbol{V}_{M_0}$ respectively. Formally, let $\hat{s}_1 \geq \hat{s}_2 \cdots \geq \hat{s}_b$ denote the singular values of $\boldsymbol{W}_1$, and let $s_1 \geq s_2 \cdots \geq s_b$ denote the singular values of $\boldsymbol{M}_0$. Let $\hat{\boldsymbol{u}}_j \in \mathbb{R}^h, \hat{\boldsymbol{v}}_j \in \mathbb{R}^d$ represent the left and right singular vectors of $\boldsymbol{W}_1$ corresponding to singular value $\hat{s}_j$. Similarly, let $\boldsymbol{u}_k \in \mathbb{R}^h, \boldsymbol{v}_k \in \mathbb{R}^d$ represent the left and right singular vectors of $\boldsymbol{M}_0$ corresponding to singular value $s_k$. Owing to the rotational invariant nature of the Gaussian matrix $\boldsymbol{W}_0$, the alignment values $\mathbb{E}\left[(\hat{\boldsymbol{u}}_j^\top \boldsymbol{u}_k)^2\right], \mathbb{E}\left[(\hat{\boldsymbol{v}}_j^\top \boldsymbol{v}_k)^2\right], \forall j, k \in \{1, \cdots, b\}$ can be computed solely based on $\hat{s}_j, s_k$ (see Landau et al. (2023); Mingo & Speicher (2017)). In this one-step context, we show that the *large $\eta$* (obtained in Section 4) leads to outlier alignment values in the overlap plots. In particular, observe from Figure 6a, Figure 6b that $(\hat{\boldsymbol{u}}_1^\top \boldsymbol{u}_1)^2, (\hat{\boldsymbol{v}}_1^\top \boldsymbol{v}_1)^2$ (i.e. alignments of top singular vectors of $\boldsymbol{W}_1, \boldsymbol{M}_0$) have high values which are close to 1. Similar observations for `GD` are presented in Appendix E. By continuing the training for 10 steps, we previously observed that the ESD transitions to an HT distribution. However, we surprisingly observed that the diagonals of the overlap matrices qualitatively exhibit an HT-like distribution as well (Figure 6c, Figure 6d). A rigorous theoretical and quantitative study is left for future work (Appendix G).

## 6  CONCLUSION

This paper presents a different angle to study the emergence of HT ESDs during NN training. Unlike existing explanations using stochastic gradient noise, we show that full-batch `GD` or `Adam` can still lead to HT ESDs in the weight matrices after only a few optimizer updates with large $\eta$. Our paper also connects with several ongoing studies in this field. In particular, our study analyzes the '5 + 1' phase model (Martin & Mahoney, 2021a) of ESD evolution and sheds light on the transitions from a Marchenko-Pastur (MP) fit (i.e. random initialization) → "Bulk+Spike" → "Bulk-Decay" → HT distributions. Our paper views the "Bulk-Decay" ESD as an intermediate state generated from diffusing the spike into the main bulk (see also Appendix E.3). Furthermore, our study tightens the connection between ESDs and feature learning, explaining why ESD-based training methods (Zhou et al., 2023) can improve the generalizability of large and deep models. Our paper also presents several surprising phenomena: (1) the emergence of the HT spectra seems to require only a single spike aligned with the teacher model; (2) the emergence of the HT spectra can appear early during training, way before the NN reaches a low training loss; (3) several factors, such as weight normalization and learning rate scheduling, can all contribute to the emergence of HT ESDs. Overall, by connecting the HT phenomenon with feature learning, we hope to promote further research into these deeply connected characteristics of NNs.

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

## A  GENERATING *Very* HEAVY-TAILED SPECTRA WITHOUT GRADIENT NOISE

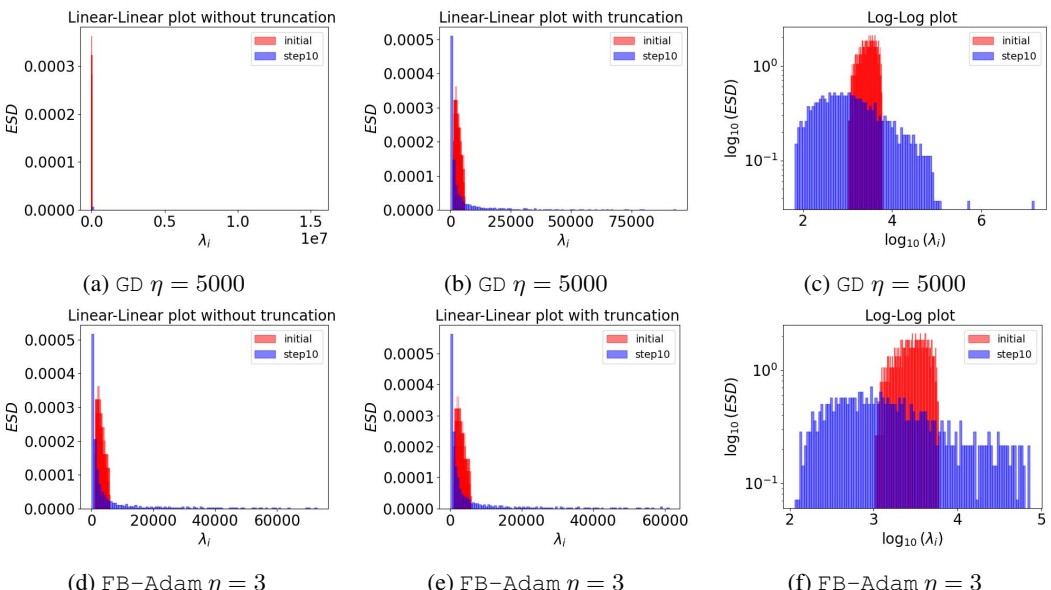

Figure 7: Emergence of HT spectra after 10 `GD`/`FB-Adam` steps with weight normalization. Here $n = 4000, d = 500, h = 3000, \sigma_* = $ `softplus`$, \sigma = $ `tanh`$, \rho_e = 0.3$. `PL_Alpha_KS` for `GD` (first row) is 1.58, and `PL_Alpha_KS` for `FB-Adam` (second row) is 1.59.

In this section, we present two examples of highly HT ESD, shown in Figure 7. If one only observes the linear-linear plot (first column), the shapes can be invisible due to large eigenvalues. For example, see Figure 7a. After truncation, they become clearer (middle column). To quantitatively verify our observations, we adopted the WeightWatcher (Martin et al., 2021) Python APIs to fit a power-law (PL) distribution to these ESDs. We find that both ESDs have a PL coefficient `PL_Alpha_KS` smaller than 2. As discussed in Martin & Mahoney (2021a, Table 3), this indicates that these ESDs have entered the *very* HT regime. More importantly, these ESDs are generated without gradient noise after only 10 `GD`/`FB-Adam` steps.

## B  PROOF OF THEOREM 4.1

In this section, we prove Theorem 4.1, and conduct numerical simulations for empirical verification. We begin by stating and discussing the main assumptions of the theorem.

**Assumption B.1** *Gaussian Initialization. The entries of the weights are sampled independently as* $[\boldsymbol{W}_0]_{ij} \overset{i.i.d.}{\sim} \mathcal{N}(0,1)$ *and* $[a]_i \overset{i.i.d.}{\sim} \mathcal{N}(0,1), \forall i \in [h], j \in [d]$.

**Assumption B.2** *Normalized Activation. The nonlinear activation $\sigma$ has $\lambda_\sigma$-bounded first three derivatives almost surely. In addition, $\sigma$ satisfies $\mathbb{E}[\sigma(z)] = 0, \mathbb{E}[z\sigma(z)] \neq 0$, for $z \sim \mathcal{N}(0,1)$.*

**Discussion.** Consider $\sigma : \mathbb{R} \to \mathbb{R}$ to be the `tanh` function and $z \sim \mathcal{N}(0,1)$. Since $\sigma(z) = $ `tanh`$(z) = \frac{e^z - e^{-z}}{e^z + e^{-z}}$, it is easy to check that $\sigma(z)$ has $1-$bounded first three derivatives and it is an odd function satisfying $\sigma(z) = -\sigma(-z)$, we have:

$$\mathbb{E}_{p(z)}[\sigma(z)] = \int_{-\infty}^{\infty} \sigma(z)p(z)\mathrm{d}z = \int_{-\infty}^{0} \sigma(z)p(z)\mathrm{d}z + \int_{0}^{\infty} \sigma(z)p(z)\mathrm{d}z = 0, \qquad (14)$$

where $p(z) = \frac{1}{\sqrt{2\pi}}e^{-\frac{z^2}{2}}$. Next, observe that

$$z\,\texttt{tanh}(z) = \frac{z(e^z - e^{-z})}{e^z + e^{-z}} \geq 0, \qquad (15)$$

where the equality $z\tanh(z) = 0$ holds only when $z = 0$. Let $\varepsilon > 0$ and expand the expectation $\mathbb{E}_{p(z)}[z\sigma(z)]$ as follows:

$$
\begin{aligned}
\mathbb{E}_{p(z)}[z\sigma(z)] &= \int_{-\infty}^{\infty} z\sigma(z)p(z)\mathrm{d}z \\
&= \int_{-\infty}^{0-\varepsilon} z\sigma(z)p(z)\mathrm{d}z + \int_{0+\varepsilon}^{\infty} z\sigma(z)p(z)\mathrm{d}z + \int_{0-\varepsilon}^{0+\varepsilon} z\sigma(z)p(z)\mathrm{d}z.
\end{aligned}
\tag{16}
$$

From equation 15, the first two terms are $> 0$, i.e:

$$
\int_{-\infty}^{0-\varepsilon} z\sigma(z)p(z)\mathrm{d}z > 0, \qquad \int_{0+\varepsilon}^{\infty} z\sigma(z)p(z)\mathrm{d}z > 0
\tag{17}
$$

whereas for $\varepsilon \to 0$, the third term can be bounded as:

$$
\int_{0-\varepsilon}^{0+\varepsilon} z\sigma(z)p(z)\mathrm{d}z \geq 0
\tag{18}
$$

By combining the above results, $\sigma$ satisfies: $\mathbb{E}_{p(z)}[z\sigma(z)] > 0 \Rightarrow \mathbb{E}_{p(z)}[z\sigma(z)] \neq 0$.

**Assumption B.3** *Teacher-Student Setup. The target labels are generated by the single index teacher model as $y_i = F^*(\boldsymbol{x}_i) + \xi_i$, where $\boldsymbol{x}_i \overset{i.i.d.}{\sim} \mathcal{N}(0, \boldsymbol{I}), \xi_i$ is i.i.d. Gaussian noise with mean 0 and variance $\rho_e^2$, and the teacher $F^*$ is $\lambda_\sigma$-Lipschitz with $\|F^*\|_{L^2} = \Theta_d(1)$.*

**Discussion.** Recall that our teacher model is given by $F^*(\boldsymbol{x}_i) = \sigma_*(\boldsymbol{\beta}^{*\top}\boldsymbol{x}_i)$, where $\boldsymbol{\beta}, \boldsymbol{x}_i \in \mathbb{R}^d, i \in [n]$, and $\sigma_*(z) = \log(1+e^z)$ is the $\texttt{softplus}$ function. Note that the derivative $\sigma_*'(z) = \frac{e^z}{1+e^z} < 1$ is bounded, and $\|\boldsymbol{\beta}^{*\top}\boldsymbol{x}_i - \boldsymbol{\beta}^{*\top}\boldsymbol{x}_j\|_2 \leq \|\boldsymbol{\beta}^*\|_2 \|\boldsymbol{x}_i - \boldsymbol{x}_j\|_2, \forall \boldsymbol{x}_i, \boldsymbol{x}_j \in \mathbb{R}^d$. This gives us:

$$
\|\sigma_*(\boldsymbol{\beta}^{*\top}\boldsymbol{x}_i) - \sigma_*(\boldsymbol{\beta}^{*\top}\boldsymbol{x}_j)\|_2 \leq \|\boldsymbol{\beta}^{*\top}\boldsymbol{x}_i - \boldsymbol{\beta}^{*\top}\boldsymbol{x}_j\|_2 \leq \|\boldsymbol{\beta}^*\|_2 \|\boldsymbol{x}_i - \boldsymbol{x}_j\|_2, \quad \forall \boldsymbol{x}_i, \boldsymbol{x}_j \in \mathbb{R}^d,
\tag{19}
$$

which implies that $F^*$ is a $\|\boldsymbol{\beta}^*\|_2$-Lipschitz function. Next, we consider $z = \boldsymbol{\beta}^{*\top}\boldsymbol{x}$, for $\boldsymbol{x} \sim \mathcal{N}(\boldsymbol{0}, \boldsymbol{I}_d)$, which implies $z \sim \mathcal{N}(0, \|\boldsymbol{\beta}^*\|_2^2)$, and bound $\sigma_*(z)$ by $0 < \sigma_*(z) < g_{\sigma_*}(z)$, where $g_{\sigma_*}(z)$ is:

$$
g_{\sigma_*}(z) = \begin{cases} 1, & z < 0 \\ z+1, & z \geq 0. \end{cases}
\tag{20}
$$

Based on these results, we calculate $\|F^*\|_{L^2}$ as follows.

$$
\|F^*\|_{L^2}^2 = \int_{\mathbb{R}} \sigma_*(z)^2 \mathrm{d}\mu < \int_{\mathbb{R}} g_{\sigma_*}(z)^2 \mathrm{d}\mu
\tag{21}
$$

where $\mathrm{d}\mu = \frac{1}{\sqrt{2\pi}\|\boldsymbol{\beta}^*\|_2} e^{-\frac{z^2}{2\|\boldsymbol{\beta}^*\|_2^2}} \mathrm{d}z$ is the gaussian measure. Further expansion of the upper bound gives:

$$
\begin{aligned}
\int_{\mathbb{R}} g_{\sigma_*}(z)^2 \mathrm{d}\mu &= \int_{\mathbb{R}} g_{\sigma_*}(z)^2 \frac{1}{\sqrt{2\pi}\|\boldsymbol{\beta}^*\|_2} e^{-\frac{z^2}{2\|\boldsymbol{\beta}^*\|_2^2}} \mathrm{d}z \\
&= \int_{-\infty}^{0} \frac{1}{\sqrt{2\pi}\|\boldsymbol{\beta}^*\|_2} e^{-\frac{z^2}{2\|\boldsymbol{\beta}^*\|_2^2}} \mathrm{d}z + \int_{0}^{+\infty} (z^2 + 2z + 1) \frac{1}{\sqrt{2\pi}\|\boldsymbol{\beta}^*\|_2} e^{-\frac{z^2}{2\|\boldsymbol{\beta}^*\|_2^2}} \mathrm{d}z \\
&= 1 + \frac{1}{2}\|\boldsymbol{\beta}^*\|_2^2 + \int_{-\infty}^{+\infty} |z| \frac{1}{\sqrt{2\pi}\|\boldsymbol{\beta}^*\|_2} e^{-\frac{z^2}{2\|\beta\|_2^2}} dz \\
&= 1 + \frac{1}{2}\|\boldsymbol{\beta}^*\|_2^2 + \sqrt{\frac{2}{\pi}}\|\boldsymbol{\beta}^*\|_2 < \infty.
\end{aligned}
$$

Since $\boldsymbol{\beta}^* \in \mathbb{S}^{d-1}$, we get: $0 < \|F^*\|_{L^2}^2 < 1 + \frac{1}{2}\|\boldsymbol{\beta}^*\|_2^2 + \sqrt{\frac{2}{\pi}}\|\boldsymbol{\beta}^*\|_2$, and $\|F^*\|_{L^2} = \Theta_d(1)$.

### B.1 NORMS OF ONE-STEP UPDATE MATRIX

We begin by formulating the full-batch gradient for the first step ($G_0$) as follows:

$$
\boldsymbol{G}_0 = \frac{1}{n\sqrt{d}} \left[ \frac{1}{\sqrt{h}} \left( \boldsymbol{a}\boldsymbol{y}^\top - \frac{1}{\sqrt{h}} \boldsymbol{a}\boldsymbol{a}^\top \sigma \left( \frac{1}{\sqrt{d}} \boldsymbol{W}_0 \boldsymbol{X}^\top \right) \right) \odot \sigma' \left( \frac{1}{\sqrt{d}} \boldsymbol{X} \boldsymbol{W}_0^\top \right) \right] \boldsymbol{X}
$$

$$
\boldsymbol{G}_0 = \underbrace{\frac{1}{n} \cdot \frac{\mu_1}{\sqrt{hd}} \boldsymbol{a}\boldsymbol{y}^\top \boldsymbol{X}}_{\boldsymbol{A}} + \frac{1}{n} \cdot \frac{1}{\sqrt{hd}} \left( \boldsymbol{a}\boldsymbol{y}^\top \odot \sigma'_\perp \left( \frac{1}{\sqrt{d}} \boldsymbol{X} \boldsymbol{W}_0^\top \right) \right) \boldsymbol{X}
$$

$$
- \frac{1}{n} \cdot \frac{1}{h\sqrt{d}} \left( \boldsymbol{a}\boldsymbol{a}^\top \sigma \left( \frac{1}{\sqrt{d}} \boldsymbol{W}_0 \boldsymbol{X}^\top \right)^\top \odot \sigma' \left( \frac{1}{\sqrt{d}} \boldsymbol{X} \boldsymbol{W}_0^\top \right) \right) \boldsymbol{X}.
\tag{22}
$$

Here we utilized the orthogonal decomposition of the activation function: $\sigma'(z) = \mu_1 + \sigma'_\perp(z)$ to the second equality. Due to Stein's lemma (Stein, 1981), we know that $\mathbb{E}[z\sigma(z)] = \mathbb{E}\left[\sigma'(z)\right] = \mu_1$, and hence $\mathbb{E}\left[\sigma'_\perp(z)\right] = 0$ for $z \sim \mathcal{N}(0,1)$.

**Lemma B.4** *((Ba et al., 2022)): Given Assumptions B.1,B.2, and B.3, let $\boldsymbol{G}_0 = \frac{1}{\eta}\left(\boldsymbol{W}_0 - \boldsymbol{W}_1\right)$ and $\boldsymbol{A} := \frac{1}{n} \cdot \frac{\mu_1}{\sqrt{hd}} \boldsymbol{a}\boldsymbol{y}^\top \boldsymbol{X}$. Then there exists a constant $c$, such that for sufficiently large $n$:*

$$
\mathbb{P}\left( \|\boldsymbol{G}_0 - \boldsymbol{A}\|_2 \le \frac{2\log^2 n}{\sqrt{n}} \|\boldsymbol{G}_0\|_2 \right) \ge 1 - ne^{-c\log^2 n} - e^{-cn}.
\tag{23}
$$

This lemma implies that $\boldsymbol{G}_0$ can be approximated by a rank one matrix $\boldsymbol{A}$ under the operator norm. Now, to analyze the FB-Adam update, recall from equation 9 that:

$$
\widetilde{\boldsymbol{G}}_0 = \frac{1 - \beta_1}{\sqrt{1 - \beta_2}} \operatorname{sign}(\boldsymbol{G}_0).
\tag{24}
$$

Observe that the essence of the first step FB-Adam update lies in the sign matrix $\operatorname{sign}(\boldsymbol{G}_0)$. Based on the expansion of $\boldsymbol{G}_0$, we leverage the rank-1 approximation matrix $\boldsymbol{A}$ to state the following lemma.

**Lemma B.5** *Given $\boldsymbol{G}_0 = \frac{1}{\eta}\left(\boldsymbol{W}_1 - \boldsymbol{W}_0\right)$ and a rank-1 matrix $\boldsymbol{A} := \frac{1}{n} \cdot \frac{\mu_1}{\sqrt{hd}} \boldsymbol{a}\boldsymbol{y}^\top \boldsymbol{X}$, then for sufficiently large $n$:*

$$
\|\operatorname{sign}(\boldsymbol{A}) - \operatorname{sign}(\boldsymbol{G}_0)\|_2 = 0, \quad \operatorname{sign}(\boldsymbol{A}) = \operatorname{sign}(\boldsymbol{G}_0) \quad \text{almost surely}
\tag{25}
$$

**Proof of Lemma B.5.** Let $a_{min} = \min_{i>0,j>0} |[\boldsymbol{A}]_{ij}|$. Since our analysis is based on large (fixed) $h$, from Lemma B.4, we almost surely have:

$$
\forall \delta > 0, \exists k > 0, \forall n > k, \|\boldsymbol{G}_0 - \boldsymbol{A}\|_2 < \delta
$$

$$
\implies \forall \delta > 0, \exists k > 0, \forall n > k, \|\boldsymbol{G}_0 - \boldsymbol{A}\|_F \le \min\{\sqrt{h}, \sqrt{d}\} \|\boldsymbol{G}_0 - \boldsymbol{A}\|_2 \le \min\{\sqrt{h}, \sqrt{d}\}\delta
\tag{26}
$$

Considering $\delta = \frac{\sqrt{a_{min}}}{\sqrt{2}\cdot\min\{\sqrt{h}, \sqrt{d}\}}$ gives us:

$$
|[\boldsymbol{G}_0]_{ij} - [\boldsymbol{A}]_{ij}|^2 < \|\boldsymbol{G}_0 - \boldsymbol{A}\|_F^2 \le \frac{a_{min}}{2},
\tag{27}
$$

which implies $\exists k > 0$, such that $\forall n > k$:

$$
\operatorname{sign}([\boldsymbol{G}_0]_{ij}) = \operatorname{sign}([\boldsymbol{A}]_{ij})
\tag{28}
$$

Thus, $\|\operatorname{sign}(\boldsymbol{A}) - \operatorname{sign}(\boldsymbol{G}_0)\|_2 = 0$, $\operatorname{sign}(\boldsymbol{A}) = \operatorname{sign}(\boldsymbol{G}_0)$.

Next, we show that every entry of the matrix $\boldsymbol{A}$ is not exactly 0 almost surely.

**Proposition B.6** *Let $\boldsymbol{A} := \frac{1}{n} \cdot \frac{\mu_1}{\sqrt{hd}} \boldsymbol{a}\boldsymbol{y}^\top \boldsymbol{X} \in \mathbb{R}^{h \times d}$, then $A_{i,j} \ne 0, \forall i \in [h], j \in [d]$ almost surely.*

**Definition B.7** *Given two measurable spaces $(\Omega, \mathcal{M}, \mu)$, $(\Omega, \mathcal{M}, \nu)$, we say $\nu$ is absolutely contin-uous with respect to $\mu$ if and only if*

$$\mu(\boldsymbol{B}) = 0 \Rightarrow \nu(\boldsymbol{B}) = 0, \quad \forall \boldsymbol{B} \in \mathcal{M}.$$

*We denote it as $\nu \ll \mu$.*

**Lemma B.8** *(Moran, 1984) Given two measurable space $(\mathbb{R}^n, \mathcal{B}(\mathbb{R}^n), \mathbf{m}^n)$, $(\mathbb{R}^n, \mathcal{B}(\mathbb{R}^n), \mathcal{L}^n)$, where $\mathbf{m}^n$ is the gaussian measure, $\mathcal{L}^n$ is the lebesgue measure, we have $\mathbf{m}^n \ll \mathcal{L}^n$.*

By *Radon-Nikodym Theorem* (Moran, 1984), we can define the Gaussian measure using the Lebesgue integral:

$$\mathcal{L}^n(E) = \int_E \mathrm{d}\mathcal{L}^n; \ \mathbf{m}^n(E) = \int_E \mathrm{d}\mathbf{m}^n = \int_E \Phi \mathrm{d}\mathcal{L}^n, \quad \forall E \in \mathcal{B}(\mathbb{R}^n), \tag{29}$$

where $\Phi$ is probability density function of $\mathbf{m}^n$ respect to $\mathcal{L}^n$. In the problem we consider, there are two groups of Gaussian measures: $a_i$ and $(X_{11}, X_{12}, ..., X_{nd}, \xi_1, ..., \xi_n)$, which induce $(\mathbb{R}, \mathcal{B}(\mathbb{R}), \mathbf{m})$ and $(\mathbb{R}^{nd+n}, \mathcal{B}(\mathbb{R}^{nd+n}), \mathbf{m}^{nd+n})$ respectively. Here $a_i$ is the $i_{th}$ element of $\boldsymbol{a}$; $X_{ij}$ is the element of $\boldsymbol{X}$; $\xi_i$ is the gaussian noise random variable.

**Proof of Proposition B.6.** Consider $\boldsymbol{A} := \frac{1}{n} \cdot \frac{\mu_1}{\sqrt{hd}} \boldsymbol{a} \boldsymbol{y}^\top \boldsymbol{X} \in \mathbb{R}^{h \times d}$, if we can prove the following:

$$\forall i \in [h], j \in [d], \mathbb{P}(A_{i,j} = 0) = 0,$$

we can further have $\mathbb{P}(\exists i, j, \text{s.t } A_{i,j} = 0) \le \sum_{i=1}^h \sum_{j=1}^d \mathbb{P}(A_{i,j} = 0) = 0$, which means $A_{i,j} \neq 0, \forall i \in [h], j \in [d]$ almost surely. So our goal is to prove $\forall i \in [h], j \in [d], \mathbb{P}(A_{i,j} = 0) = 0$. Given $\boldsymbol{a} \in \mathbb{R}^{h \times 1}, \boldsymbol{y}^\top \boldsymbol{X} \in \mathbb{R}^{1 \times d}$, notice that

$$\{A_{i,j} = 0\} \Leftrightarrow \{a_i = 0\} \bigcup \{(y^\top X)_j = 0\}.$$

Since $\boldsymbol{a}$ and $\boldsymbol{y}^\top \boldsymbol{X}$ are independent, we aim to prove $\forall i \in [h], j \in [d], \mathbf{m}^{nd+n}((y^\top X)_j = 0) = 0$ and $\mathbf{m}(a_i = 0) = 0$.

**We first show** $\forall j \in [d], \mathbb{P}((y^\top X)_j = 0) = 0$. Consider $y = \sigma_*(\boldsymbol{X}\boldsymbol{\beta}^*) + \xi$, where $\sigma_*$ is a `Softplus` function and $\boldsymbol{\beta}^* = (b_1, ..., b_d)^\top$ to get:

$$(y^\top X)_j = \sum_{i=1}^n X_{ij} \left[ \ln \left( \exp \left( \sum_{k=1}^d b_k X_{ik} \right) + 1 \right) + \xi_i \right] \tag{30}$$

It is easy to observe that $(y^\top X)_j$ can be written as a function:

$$(y^\top X)_j = f_j(X_{11}, \cdots, X_{1d}, \cdots, X_{nd}, \xi_1, \cdots, \xi_n). \tag{31}$$

We can easily verify $f_j : \mathbb{R}^{nd+n} \to \mathbb{R}$ is continuously differentiable of the first order, and we denote it as $f_j \in \mathbf{C}^1(\mathbb{R}^{nd+n})$. Considering the set

$$\mathcal{M}_1 = \left\{ (X_{11}, \cdots, X_{1d}, \cdots, X_{nd}, \xi_1, \cdots, \xi_n) \in \mathbb{R}^{nd+n} \right. \\ \left. | f_j(X_{11}, \cdots, X_{1d}, \cdots, X_{nd}, \xi_1, \cdots, \xi_n) = 0 \right\}, \tag{32}$$

due to $f_j \in \mathbf{C}^1(\mathbb{R}^{nd+n})$ and $\text{rank}(\mathbf{D}f_j) = \text{rank}(\frac{\partial f_j}{\partial X_{11}}, \cdots, \frac{\partial f_j}{\partial X_{1d}}, \cdots, \frac{\partial f_j}{\partial X_{nd}}, \frac{\partial f_j}{\partial \xi_1}, \cdots, \frac{\partial f_j}{\partial \xi_n}) = 1$, by *Implicit Function Theorem* (Zorich & Paniagua, 2016), we have $\mathcal{M}_1$ is a $C^1$ $(nd + n - 1)$-dim sub-manifold. Therefore $\mathcal{L}^{nd+n}(\mathcal{M}_1) = \int \mathcal{M}_1 \mathrm{d}\mathcal{L}^{nd+n} = 0$.

• By Lemma B.8, we have $\mathbf{m}^{nd+n} \ll \mathcal{L}^{nd+n}$. Then $\mathbf{m}^{nd+n}(\mathcal{M}_1) = 0$, we can get $\forall j \in [d], \mathbf{m}^{nd+n}((y^\top X)_j = 0) = 0$.

• Observe that since any single point set is a zero-measure set for Lebesgue measure $\mathcal{L}^1$ and $\mathbf{m} \ll \mathcal{L}^1$, we get $\forall i \in [h], \mathbf{m}(a_i = 0) = 0$.

Since we have proved $\forall i \in [h], j \in [d], \mathbf{m}^{nd+n}((y^\top X)_j = 0) = 0$ and $\mathbf{m}(a_i = 0) = 0$, we get $\forall i \in [h], j \in [d], \mathbb{P}(A_{i,j} = 0) = 0$. Thus proving the proposition.

**Lemma B.9** *(Forster et al., 2001) Let $M \in \{-1, +1\}^{h \times d}$ and $M' \in \mathbb{R}^{h \times d}$ such that* $\text{sign}\left(M_{i,j}\right) = \text{sign}\left(M'_{i,j}\right)$ *for all $i \in [h], j \in [d]$. Then the following holds:*

$$\text{rank}\left(M'\right) \geq \frac{\sqrt{hd}}{\|M\|_2}. \tag{33}$$

From Lemma B.5 and Proposition B.6 it is clear that $\text{sign}(G_0)$ almost surely only contains $\{-1, 1\}$. Now, by combining Lemma B.9 and Lemma B.4 for sufficiently large $n$, almost surely leads to:

$$\text{rank}(A) = 1 \geq \frac{\sqrt{hd}}{\|\text{sign}(A)\|_2} \implies 1 \geq \frac{\sqrt{hd}}{\|\text{sign}(G_0)\|_2} \tag{34}$$

Therefore, we have:

$$\left\|\widetilde{G}_0\right\|_2 = \Omega_{d,\mathbb{P}}(\sqrt{hd}) \tag{35}$$

Additionally:

$$\left\|\widetilde{G}_0\right\|_2 \leq \left\|\widetilde{G}_0\right\|_F = \left\|\frac{1-\beta_1}{\sqrt{1-\beta_2}} \text{sign}(G_0)\right\|_F = \frac{1-\beta_1}{\sqrt{1-\beta_2}}\sqrt{hd}$$
$$\implies \left\|\widetilde{G}_0\right\|_2 = O(\sqrt{hd}). \tag{36}$$

Finally, by combined equations 36 and 35, we get:

$$\left\|\widetilde{G}_0\right\|_2 = \Theta_{d,\mathbb{P}}(\sqrt{hd}), \quad \left\|\widetilde{G}_0\right\|_F = \Theta_{d,\mathbb{P}}(\sqrt{hd}). \tag{37}$$

Thus proving the theorem.

## B.2 NUMERICAL SIMULATIONS

We consider multiple sets of $n, h, d$ (see Table 2) for one-step `FB-Adam` and plot the Frobenius norm and spectral norm of $\widetilde{G}_0$. Figure 8 shows a linear relationship of the norms with $\sqrt{hd}$, which validates the results in our theorem: $\left\|\widetilde{G}_0\right\|_2 = \Theta_{d,\mathbb{P}}(\sqrt{hd}), \quad \left\|\widetilde{G}_0\right\|_F = \Theta_{d,\mathbb{P}}(\sqrt{hd}).$

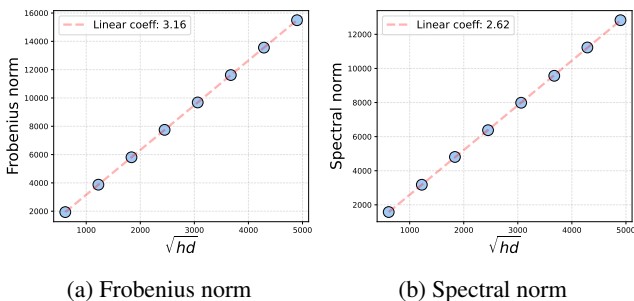

(a) Frobenius norm        (b) Spectral norm

Figure 8: Plots of $\left\|\widetilde{G}_0\right\|_F, \left\|\widetilde{G}_0\right\|_2$ with varying $n, d, h$ and $\beta_1 = 0.9, \beta_2 = 0.999$.

## C DISCUSSIONS ON MEAN-FIELD INITIALIZATION

Our setup employs the widely studied NTK initialization (Wang et al., 2023) for the two-layer NNs. Alternatively, previous studies have also focused on mean-field-based initialization (Ba et al., 2022; Moniri et al., 2023) to analyze the role of one-step optimizer updates. The mean-field initialization for a two-layer NN can be formulated as:

$$f(x_i) = \frac{1}{\sqrt{h}} a^\top \sigma\left(W x_i\right). \tag{38}$$

| Index | $n$ | $h$ | $d$ | Optimizer |
|-------|------|------|------|-----------|
| 0 | 1000 | 750 | 500 | FB-Adam |
| 1 | 2000 | 1500 | 1000 | FB-Adam |
| 2 | 3000 | 2250 | 1500 | FB-Adam |
| 3 | 4000 | 3000 | 2000 | FB-Adam |
| 4 | 5000 | 3750 | 2500 | FB-Adam |
| 5 | 6000 | 4500 | 3000 | FB-Adam |
| 6 | 7000 | 5250 | 3500 | FB-Adam |
| 7 | 8000 | 6000 | 4000 | FB-Adam |

Table 2: Parameters for Figure 8

Here $\boldsymbol{W} \in \mathbb{R}^{h \times d}, \boldsymbol{a} \in \mathbb{R}^h$ are the first and second layer weights respectively, with entries sampled as $[\boldsymbol{W}_0]_{ij} \overset{\text{i.i.d.}}{\sim} \mathcal{N}(0, \frac{1}{d}), [a]_i \overset{\text{i.i.d.}}{\sim} \mathcal{N}(0, \frac{1}{h}), \forall i \in [h], j \in [d]$. Notice the change in scale of the entries and the $\frac{1}{\sqrt{h}}$ scaling factor in equation 38. In this setup: $\|\boldsymbol{W}_0\|_2 = \Theta_{d,\mathbb{P}}(1), \|\boldsymbol{W}_0\|_F = \Theta_{d,\mathbb{P}}(\sqrt{h})$.

This section provides additional results and discussions for the mean-field initialization. We claim that it is straightforward to extend the conclusions regarding the scale of $\eta$ for one-step FB-Adam to this setting. In particular, with *large* $\eta$ and multiple optimizer steps, we observe the emergence of HT ESDs. Additionally, the alignments of singular vectors of the weight and corresponding optimizer update matrices also remain a potential contributor to the emergence of HT ESDs, consistent with the discussion in the main paper.

## C.1 SCALING $\eta$ AND ALIGNMENT OF SINGULAR VECTORS FOR FB-ADAM

**A note on notation from Ba et al. (2022):** In our setup, we denote $\boldsymbol{G}_0 = \frac{1}{\eta}(\boldsymbol{W}_1 - \boldsymbol{W}_0)$, whereas Ba et al. (2022) consider $\boldsymbol{G}_0 = \frac{1}{\eta\sqrt{h}}(\boldsymbol{W}_1 - \boldsymbol{W}_0)$. Thus, in the mean-field setting, the learning rates we obtain are simply the scaled versions of theirs by a factor of $\sqrt{h}$.

To this end, Ba et al. (2022) showed that $\eta = \Theta(h)$ (scaling adjusted to our notation) is (sufficiently) large for the GD update $\boldsymbol{G}_0$ (see Figure 9). One can also verify that the results of Theorem 4.1 for FB-Adam can be scaled and extended to this setting.

**Corollary C.1** *Under the assumptions of Theorem 4.1, we obtain the following scaling for $\eta$ in the mean-field initialization setting:*

$$
\begin{aligned}
\eta = \Theta(1/\sqrt{d}) &\implies \|\boldsymbol{W}_1 - \boldsymbol{W}_0\|_F \asymp \|\boldsymbol{W}_0\|_F \\
\eta = \Theta\left(1/\sqrt{hd}\right) &\implies \|\boldsymbol{W}_1 - \boldsymbol{W}_0\|_2 \asymp \|\boldsymbol{W}_0\|_2,
\end{aligned}
\tag{39}
$$

*where $\|\boldsymbol{W}_0\|_2 = \Theta_{d,\mathbb{P}}(\sqrt{1}), \|\boldsymbol{W}_0\|_F = \Theta_{d,\mathbb{P}}(\sqrt{h})$.*

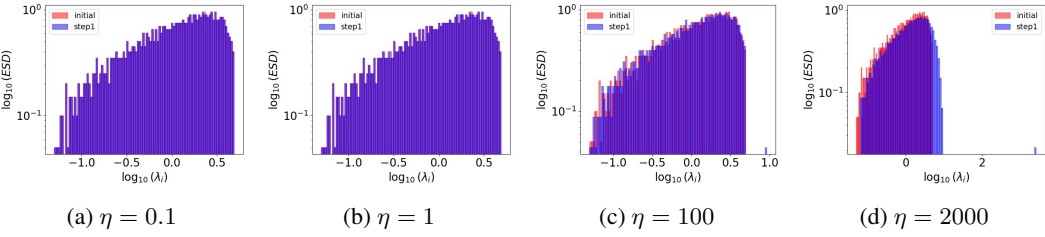

(a) $\eta = 0.1$        (b) $\eta = 1$        (c) $\eta = 100$        (d) $\eta = 2000$

Figure 9: Evolution of ESD of $\boldsymbol{W}^\top \boldsymbol{W}$ after one step GD optimizer update in mean-field setting. Here $n = 2000, d = 1000, h = 1500, \sigma_* = \text{softplus}, \sigma = \text{tanh}, \rho_e = 0.3$.

**Singular Vector Overlaps after one-step FB-Adam update.** Following the main paper, we compute the following overlap metrics: $\mathcal{O}(\boldsymbol{U}_{W_0}, \boldsymbol{U}_{M_0}), \mathcal{O}(\boldsymbol{V}_{W_0}, \boldsymbol{V}_{M_0}), \mathcal{O}(\boldsymbol{U}_{W_1}, \boldsymbol{U}_{M_0}), \mathcal{O}(\boldsymbol{V}_{W_1}, \boldsymbol{V}_{M_0})$. For FB-Adam with $\eta = 0.04$ (which is a large $\eta$ in this setting), Figure 11 shows the outliers for $\mathcal{O}(\boldsymbol{U}_{W_1}, \boldsymbol{U}_{M_0}), \mathcal{O}(\boldsymbol{V}_{W_1}, \boldsymbol{V}_{M_0})$ corresponding to the spike in Figure 10c. Thus aligning with the results in Section 5.3.

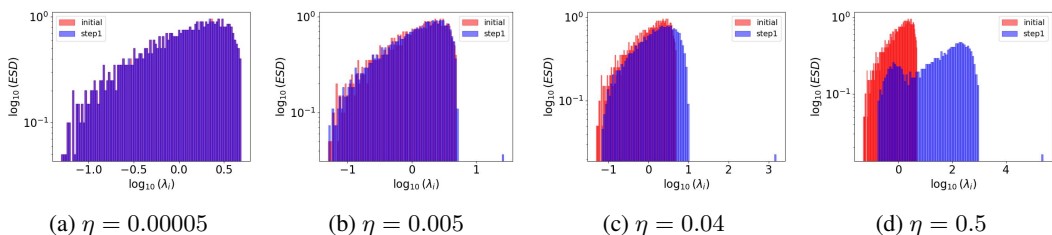

(a) $\eta = 0.00005$     (b) $\eta = 0.005$     (c) $\eta = 0.04$     (d) $\eta = 0.5$

Figure 10: Evolution of ESD of $\boldsymbol{W}^\top \boldsymbol{W}$ after one step `FB-Adam` update in mean-field setting. Here $n = 2000, d = 1000, h = 1500, \sigma_* = \texttt{softplus}, \sigma = \texttt{tanh}, \rho_e = 0.3$.

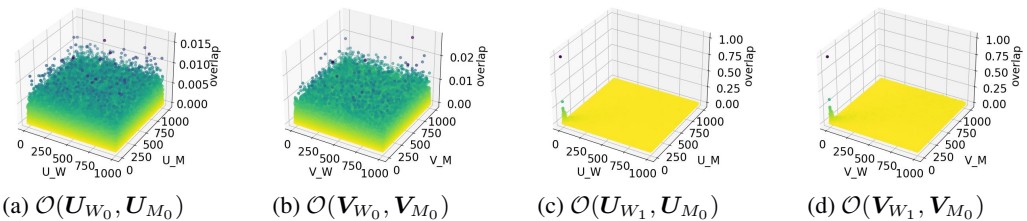

(a) $\mathcal{O}(\boldsymbol{U}_{W_0}, \boldsymbol{U}_{M_0})$    (b) $\mathcal{O}(\boldsymbol{V}_{W_0}, \boldsymbol{V}_{M_0})$    (c) $\mathcal{O}(\boldsymbol{U}_{W_1}, \boldsymbol{U}_{M_0})$    (d) $\mathcal{O}(\boldsymbol{V}_{W_1}, \boldsymbol{V}_{M_0})$

Figure 11: Overlaps after one `FB-Adam` update with $\eta = 0.04$ in mean-field setting.

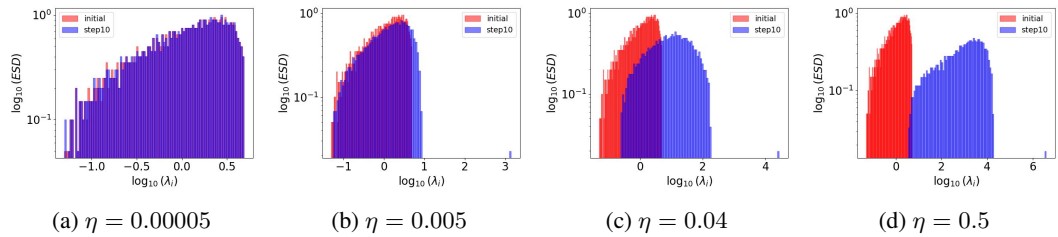

(a) $\eta = 0.00005$     (b) $\eta = 0.005$     (c) $\eta = 0.04$     (d) $\eta = 0.5$

Figure 12: Evolution of ESD of $\boldsymbol{W}^\top \boldsymbol{W}$ after 10 `FB-Adam` updates in mean-field setting. Here $n = 2000, d = 1000, h = 1500, \sigma_* = \texttt{softplus}, \sigma = \texttt{tanh}, \rho_e = 0.3$.

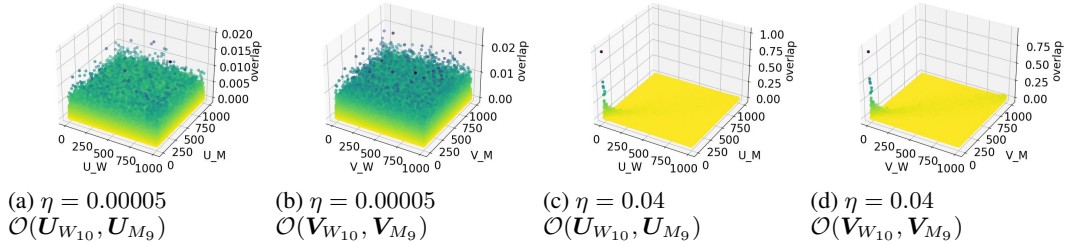

(a) $\eta = 0.00005$ $\mathcal{O}(\boldsymbol{U}_{W_{10}}, \boldsymbol{U}_{M_9})$    (b) $\eta = 0.00005$ $\mathcal{O}(\boldsymbol{V}_{W_{10}}, \boldsymbol{V}_{M_9})$    (c) $\eta = 0.04$ $\mathcal{O}(\boldsymbol{U}_{W_{10}}, \boldsymbol{U}_{M_9})$    (d) $\eta = 0.04$ $\mathcal{O}(\boldsymbol{V}_{W_{10}}, \boldsymbol{V}_{M_9})$

Figure 13: Overlaps between singular vectors after 10 `FB-Adam` updates with $\eta = 0.00005$ (plots (a), (b)) and $\eta = 0.04$ (plots (c), (d)) in the mean-field setting.

**Heavy-Tailed Phenomenon after Multiple Steps With Mean-Field Initialization.** Similar to the experiments in Section 5, we employ the mean-field initialization and apply 10 `FB-Adam` updates with various $\eta$ to compute $\mathcal{O}(\boldsymbol{U}_{W_{10}}, \boldsymbol{U}_{M_9}), \mathcal{O}(\boldsymbol{V}_{W_{10}}, \boldsymbol{V}_{M_9})$. Notice that for small $\eta = 0.00005$, the ESD is expected to remain largely unchanged and the overlap plots illustrate random alignment values. However, for $\eta = 0.04$, the outliers emerge in the overlap matrices in Figure 13 and correlate with the HT ESD in Figure 12.

## D ESD EVOLUTION OVER LONG TRAINING PERIODS

In the main text, we claimed and validated that the ESD of the hidden layer weight matrix exhibits heavy tails after multiple steps of GD/ FB-Adam with sufficiently large $\eta$. In the following ESD plots, we further justify our claim and illustrate that very long training $t = 10000$ with small $\eta$ does not result in HT ESDs.

### D.1 ESD EVOLUTION WITH GD

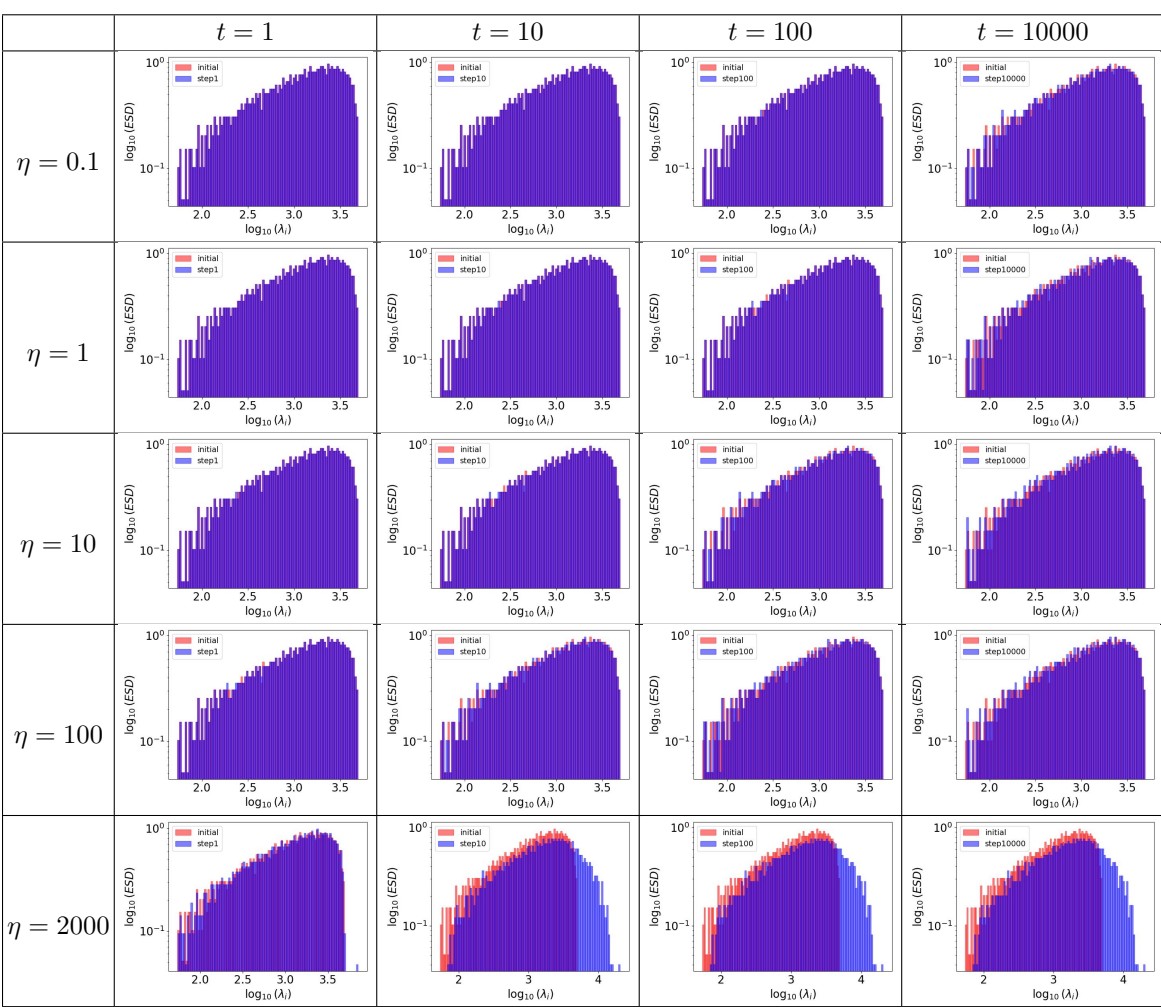

Table 3: Evolution of ESD over different step times $t \in \{1, 10, 100, 10000\}$ with different $\eta \in \{0.1, 1, 10, 100, 2000\}$ for GD optimizer. Through this grid, we highlight the critical $\eta$ for the occurrence of a spike at $t = 1$ and the effects of longer training on the emergence of HT ESD.

## D.2   ESD EVOLUTION WITH FB-ADAM

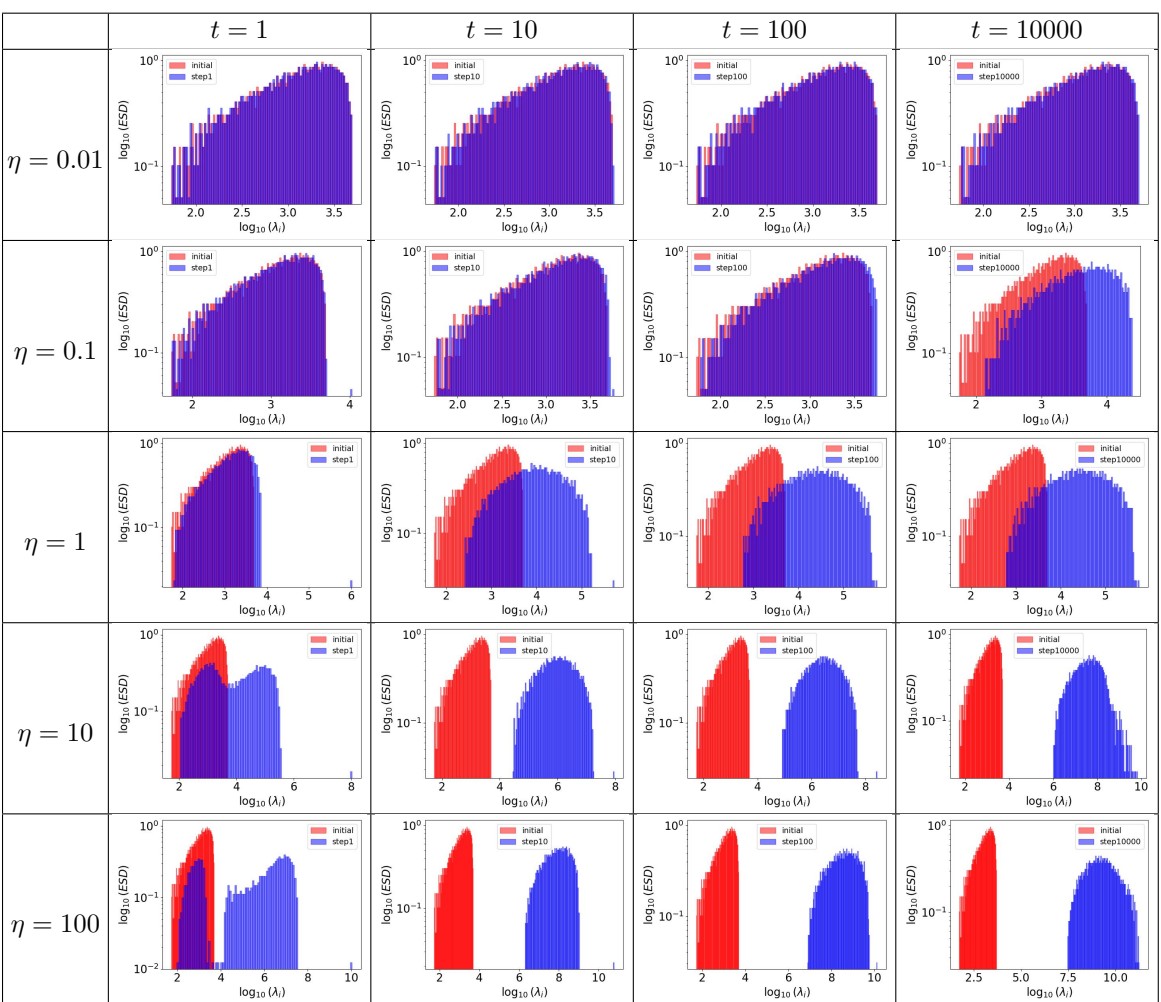

Table 4: Evolution of ESD over different step times $t \in \{1, 10, 100, 10000\}$ with different $\eta \in \{0.01, 0.1, 1, 10, 100\}$ for FB-Adam optimizer. Through this grid, we highlight the critical $\eta$ for the occurrence of a spike at $t = 1$ and the effects of longer training on the HT emergence.

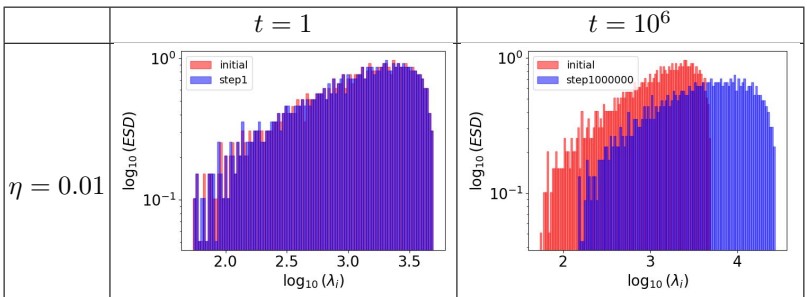

Table 5: ESD after $t = \{1, 10^6\}$ steps of FB-Adam with $\eta = 0.01$. The plot showcases a scenario where a spike does not appear after the first step but results in HT ESD after extremely long training.

# E  ADDITIONAL EXPERIMENTS

**Hyperparameters:** In most of our experiments, we follow a consistent setup with $n = 2000$, $n\_test = 200$, $d = 1000$, $h = 1500$, $\lambda = 0.01$, $\rho_e = 0.3$, $\sigma_* = \texttt{softplus}$, $\sigma = \texttt{tanh}$. Additionally, we explicitly mention the parameter changes wherever applicable in our experiments.

## E.1  ONE-STEP OPTIMIZER UPDATES

In this section, we present additional experiments for one-step optimizer updates. Figure 14 leverages the same experimental setup as Section 4 and illustrates the ESD of $\boldsymbol{W}^\top \boldsymbol{W}$ after the first step of $\texttt{GD}$ and $\texttt{FB-Adam}$. Based on the results obtained in the main text, $\eta = 2000$ is an extremely large learning rate for $\texttt{FB-Adam}$, which results in a clear bimodal distribution. Note that the tendency towards such a distribution was already observed with $\eta = 10$ in Figure 2 (Section 4). Based on the same setup as Section 5.3, Figures 15, 16, 17 represent the overlaps of singular vectors after one-step update and showcase the presence of outliers for sufficiently large $\eta$. Finally, we illustrate the role of sample sizes on the losses, $\texttt{KTA}$ and $\texttt{sim}(\boldsymbol{W}, \boldsymbol{\beta}^*)$ in Figure 18 (for $n = 4000$) and in Figure 19 (for $n = 8000$).

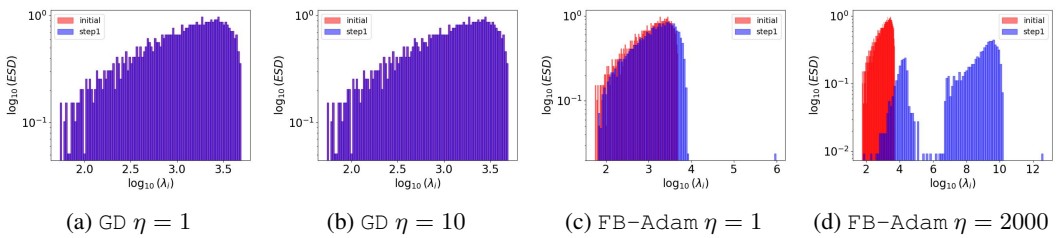

| (a) $\texttt{GD}\ \eta = 1$ | (b) $\texttt{GD}\ \eta = 10$ | (c) $\texttt{FB-Adam}\ \eta = 1$ | (d) $\texttt{FB-Adam}\ \eta = 2000$ |

Figure 14: Evolution of ESD of $\boldsymbol{W}^\top \boldsymbol{W}$ after one step optimizer update with varying learning rates. Here $n = 2000$, $d = 1000$, $h = 1500$, $\sigma_* = \texttt{softplus}$, $\sigma = \texttt{tanh}$, $\rho_e = 0.3$.

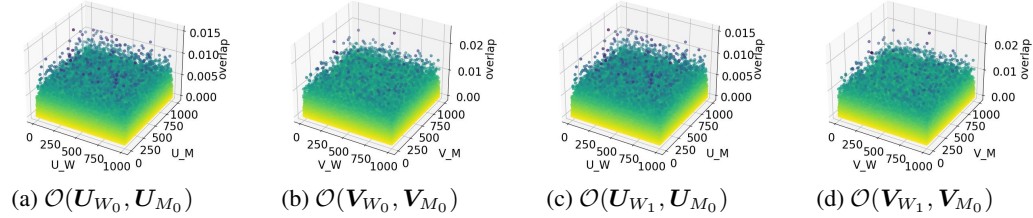

| (a) $\mathcal{O}(\boldsymbol{U}_{W_0}, \boldsymbol{U}_{M_0})$ | (b) $\mathcal{O}(\boldsymbol{V}_{W_0}, \boldsymbol{V}_{M_0})$ | (c) $\mathcal{O}(\boldsymbol{U}_{W_1}, \boldsymbol{U}_{M_0})$ | (d) $\mathcal{O}(\boldsymbol{V}_{W_1}, \boldsymbol{V}_{M_0})$ |

Figure 15: Overlaps between singular vectors after one step $\texttt{GD}$ update with $\eta = 0.1$. Here $n = 2000$, $d = 1000$, $h = 1500$, $\sigma_* = \texttt{softplus}$, $\sigma = \texttt{tanh}$, $\rho_e = 0.3$.

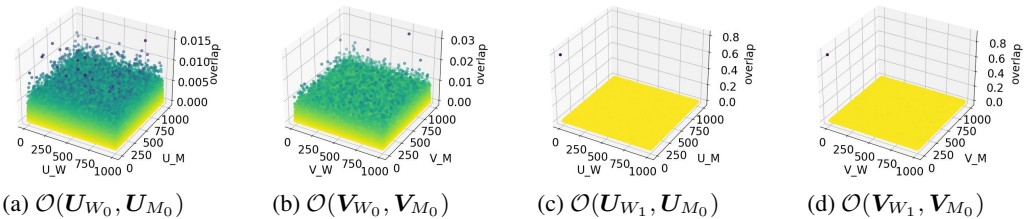

| (a) $\mathcal{O}(\boldsymbol{U}_{W_0}, \boldsymbol{U}_{M_0})$ | (b) $\mathcal{O}(\boldsymbol{V}_{W_0}, \boldsymbol{V}_{M_0})$ | (c) $\mathcal{O}(\boldsymbol{U}_{W_1}, \boldsymbol{U}_{M_0})$ | (d) $\mathcal{O}(\boldsymbol{V}_{W_1}, \boldsymbol{V}_{M_0})$ |

Figure 16: Overlaps between singular vectors after one step $\texttt{FB-Adam}$ update with $\eta = 0.1$. Here $n = 2000$, $d = 1000$, $h = 1500$, $\sigma_* = \texttt{softplus}$, $\sigma = \texttt{tanh}$, $\rho_e = 0.3$.

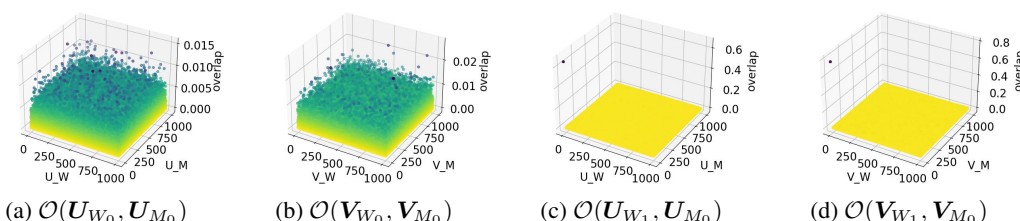

(a) $\mathcal{O}(\boldsymbol{U}_{W_0}, \boldsymbol{U}_{M_0})$     (b) $\mathcal{O}(\boldsymbol{V}_{W_0}, \boldsymbol{V}_{M_0})$     (c) $\mathcal{O}(\boldsymbol{U}_{W_1}, \boldsymbol{U}_{M_0})$     (d) $\mathcal{O}(\boldsymbol{V}_{W_1}, \boldsymbol{V}_{M_0})$

Figure 17: Overlaps between singular vectors after one step GD update with $\eta = 2000$. Here $n = 2000, d = 1000, h = 1500, \sigma_* = \texttt{softplus}, \sigma = \texttt{tanh}, \rho_e = 0.3$.

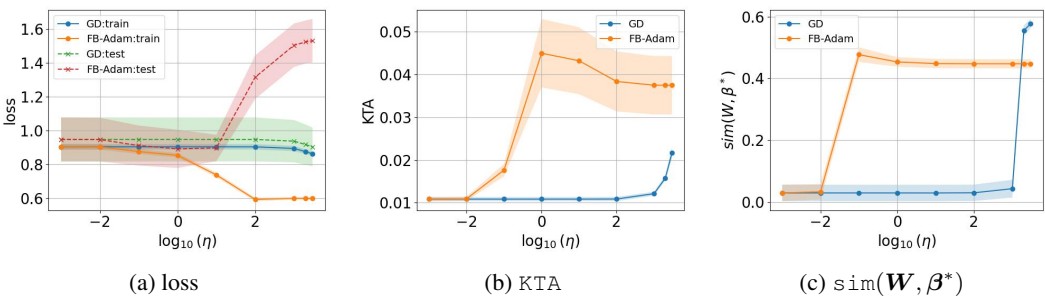

(a) loss           (b) KTA          (c) $\texttt{sim}(\boldsymbol{W}, \boldsymbol{\beta}^*)$

Figure 18: Train/test losses, KTA, $\texttt{sim}(\boldsymbol{W}, \boldsymbol{\beta}^*)$ for $f(\cdot)$ trained with one-step of GD, FB-Adam. Here $n = 4000, d = 1000, h = 1500, \sigma_* = \texttt{softplus}, \sigma = \texttt{tanh}, \rho_e = 0.3$.

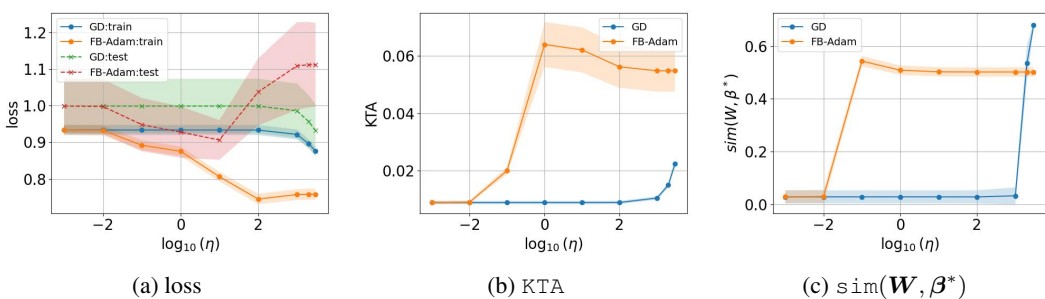

(a) loss           (b) KTA          (c) $\texttt{sim}(\boldsymbol{W}, \boldsymbol{\beta}^*)$

Figure 19: Train/test losses, KTA, $\texttt{sim}(\boldsymbol{W}, \boldsymbol{\beta}^*)$ for $f(\cdot)$ trained with one-step of GD, FB-Adam. Here $n = 8000, d = 1000, h = 1500, \sigma_* = \texttt{softplus}, \sigma = \texttt{tanh}, \rho_e = 0.3$.

## E.2 10 STEP OPTIMIZER UPDATES

In this section, we present additional experiments for 10 optimizer updates. Figure 20 presents the losses, ESD of $W^\top W$ and the overlaps of singular vectors after 10 steps with $\texttt{GD}(\eta = 2000)$. Notice that the prominent outlier values in Figures 17c, 17d after the one-step update have now reduced significantly. Thus illustrating the varying spread of values even for the left and right singular vector overlaps. The role of $\eta$ on the losses, $\texttt{KTA}$ and the ESD metric $\texttt{PL\_Alpha\_Hill}$ are illustrated in Figures 21, 22. We illustrate the ESD of $W^\top W$ after 10 steps with $n = 8000$ and learning rates $\eta$ chosen from $\{1, 10, 100, 1000\}$ in Figure 23. Observe that as $\eta$ increases, the spike tends to move far away from the bulk and significantly distorts the shape of the bulk only for $\eta = 1000$. Finally, in Figures 24, 25 we illustrate the role of label noise increasing from $\rho_e = 0.3$ (Figure 24) to $\rho_e = 0.7$ (Figure 25). Although the ESDs look the same in both cases, note that the outlier (max) values of the overlap matrices for $\rho_e = 0.7$ have relatively larger values compared to the $\rho_e = 0.3$ case.

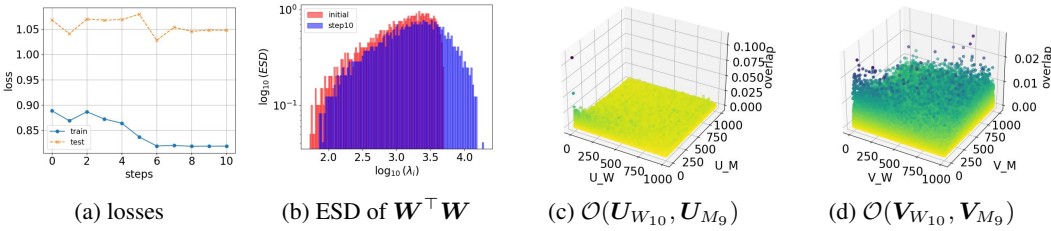

(a) losses      (b) ESD of $W^\top W$      (c) $\mathcal{O}(U_{W_{10}}, U_{M_9})$      (d) $\mathcal{O}(V_{W_{10}}, V_{M_9})$

Figure 20: Losses, ESD, and Overlaps between singular vectors after 10 steps of $\texttt{GD}(\eta = 2000)$. Here $n = 2000, d = 1000, h = 1500, \sigma_* = \texttt{softplus}, \sigma = \texttt{tanh}, \rho_e = 0.3$.

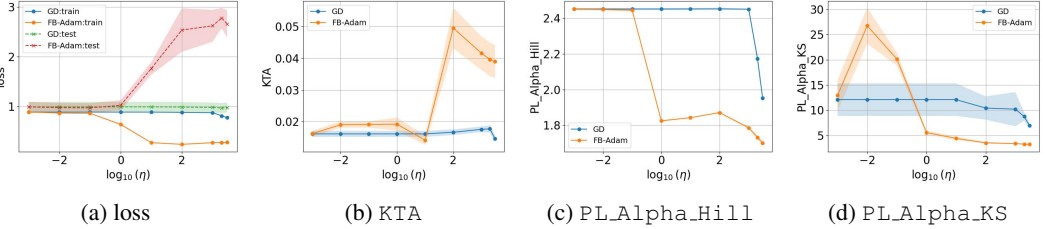

(a) loss      (b) $\texttt{KTA}$      (c) $\texttt{PL\_Alpha\_Hill}$      (d) $\texttt{PL\_Alpha\_KS}$

Figure 21: Train/test losses, $\texttt{KTA}$, $\texttt{PL\_Alpha\_Hill}$, $\texttt{PL\_Alpha\_KS}$ for $f(\cdot)$ trained with 10 steps of $\texttt{GD}, \texttt{FB-Adam}$. Here $n = 2000, d = 1000, h = 1500, \sigma_* = \texttt{softplus}, \sigma = \texttt{tanh}, \rho_e = 0.3$.

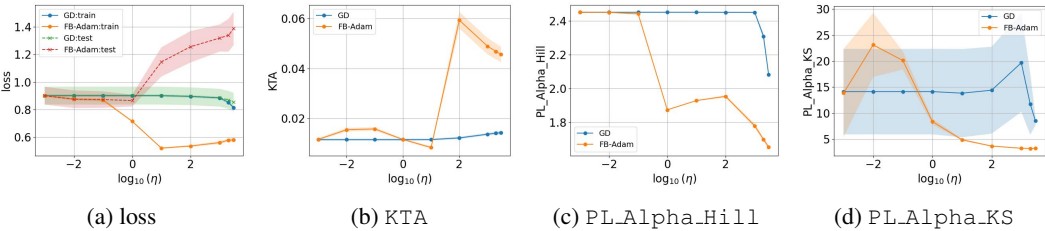

(a) loss      (b) $\texttt{KTA}$      (c) $\texttt{PL\_Alpha\_Hill}$      (d) $\texttt{PL\_Alpha\_KS}$

Figure 22: Train/test losses, $\texttt{KTA}$, $\texttt{PL\_Alpha\_Hill}$, $\texttt{PL\_Alpha\_KS}$ for $f(\cdot)$ trained with 10 steps of $\texttt{GD}, \texttt{FB-Adam}$. Here $n = 4000, d = 1000, h = 1500, \sigma_* = \texttt{softplus}, \sigma = \texttt{tanh}, \rho_e = 0.3$.

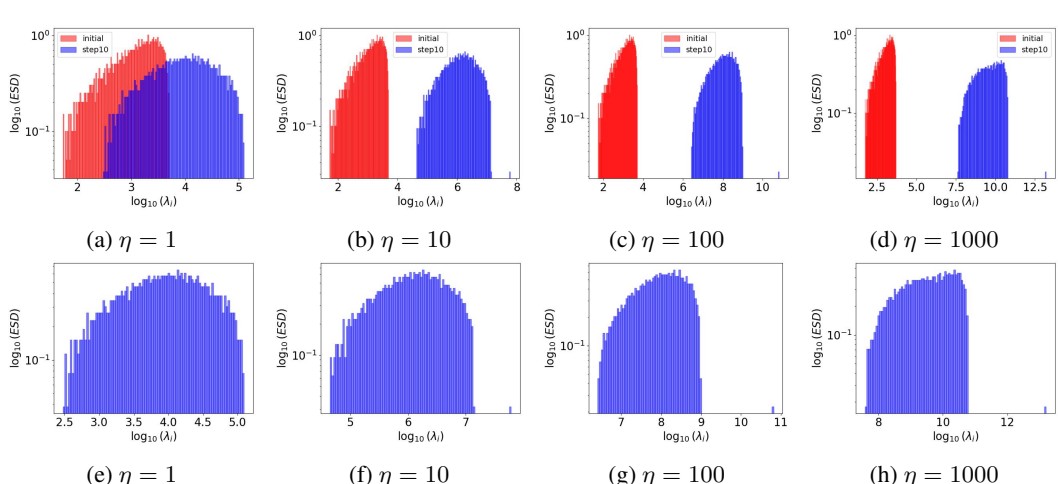

Figure 23: Evolution of ESD of $\boldsymbol{W}^\top \boldsymbol{W}$ after 10 steps of FB-Adam updates with $n = 8000, d = 1000, h = 1500, \lambda = 0.01, \rho_e = 0.3$. The first row compares the initial and final ESDs. The second row illustrates solely the final ESD of $\boldsymbol{W}^\top \boldsymbol{W}$ (i.e. $\boldsymbol{W}_{10}^\top \boldsymbol{W}_{10}$) for better visualizations of the shape.

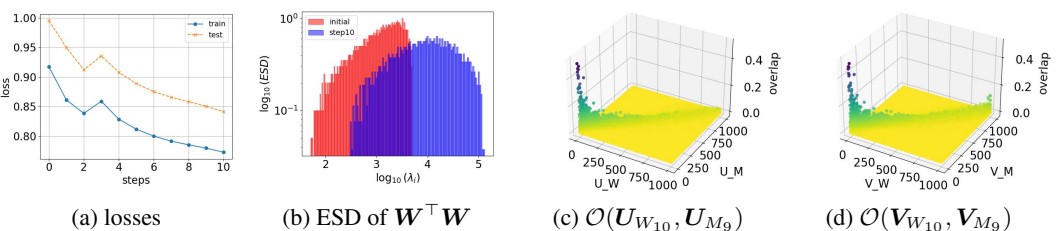

Figure 24: Losses, ESD, and Overlaps between singular vectors after 10 FB-Adam($\eta = 1$) steps for $n = 8000, d = 1000, h = 1500, \lambda = 0.01, \rho_e = 0.3$.

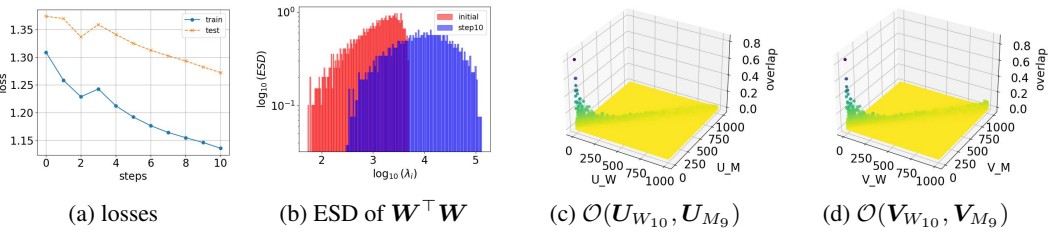

Figure 25: Losses, ESD, and Overlaps between singular vectors after 10 FB-Adam($\eta = 1$) steps for $n = 8000, d = 1000, h = 1500, \lambda = 0.01, \rho_e = 0.7$.

## E.3 SPIKE MOVEMENT WITH GD

During our experiments with GD($\eta = 2000$), we observed a surprising transition in the position of the spike relative to the bulk of the ESD. Particularly between steps 5 and 6, the spike in the ESD of $\boldsymbol{W}^\top \boldsymbol{W}$ which represents the largest singular value, reduces in value by an order of magnitude. Additionally, this reduction seems to be correlated with the reduction in maximum overlap values from $\max(\mathcal{O}(\boldsymbol{U}_{W_5}, \boldsymbol{U}_{M_5}))$ to $\max(\mathcal{O}(\boldsymbol{U}_{W_6}, \boldsymbol{U}_{M_5}))$ (see Figure 26). To understand this behavior, we emphasize the spike in the ESD of $\boldsymbol{W}_5^\top \boldsymbol{W}_5$ in Figure 26a and the large overlap value (black dot) in Figure 26b. Let $\hat{u}_{W_5}, \hat{u}_{M_5}, \hat{u}_{W_6} \in \mathbb{R}^h$ represent the left singular vectors corresponding to the largest singular values in $\boldsymbol{W}_5, \boldsymbol{M}_5, \boldsymbol{W}_6$ respectively. The large overlap value (black dot) in Figure 26b intuitively represents a high degree of overlap/alignment between $\hat{u}_{W_5}, \hat{u}_{M_5}$. As a result of obtaining $\boldsymbol{W}_6$ by $\boldsymbol{W}_6 = \boldsymbol{W}_5 + \boldsymbol{M}_5$, the singular vector $\hat{u}_{W_6}$ seems to be rotated from $\hat{u}_{W_5}$ in such a way that its alignment with $\hat{u}_{M_5}$ is reduced (see Figure 26d). One can intuitively think of this process as the 'diffusion'/'spread' of the overlap between $\hat{u}_{M_5}$ and all left singular vectors of $\boldsymbol{W}_6$.

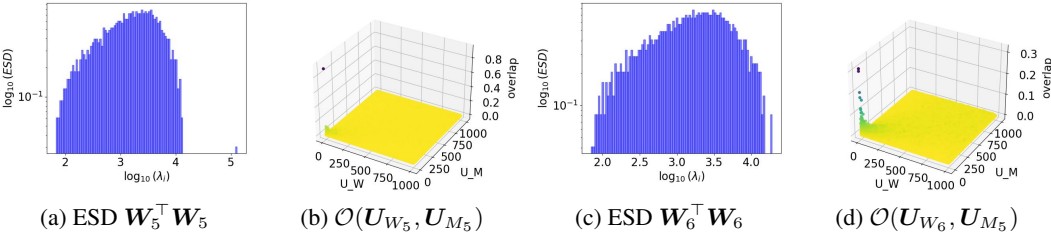

(a) ESD $\boldsymbol{W}_5^\top \boldsymbol{W}_5$     (b) $\mathcal{O}(\boldsymbol{U}_{W_5}, \boldsymbol{U}_{M_5})$     (c) ESD $\boldsymbol{W}_6^\top \boldsymbol{W}_6$     (d) $\mathcal{O}(\boldsymbol{U}_{W_6}, \boldsymbol{U}_{M_5})$

Figure 26: Phase transition in ESD of $\boldsymbol{W}^\top \boldsymbol{W}$ between steps $5, 6$ when updated using GD ($\eta = 2000$). Here $n = 2000, d = 1000, h = 1500, \sigma_* = \texttt{softplus}, \sigma = \texttt{tanh}, \rho_e = 0.3$.

## E.4 ON GENERALIZATION, LEARNING RATE SCHEDULES AND WEIGHT NORMALIZATION

**Role of sample size** ($n$). By fixing $d = 1000, h = 1500$ and $\eta = 1$, we vary the size of the training dataset $n$ as per the set $\{500, 2000, 4000, 8000\}$. In this setting, observe from Figure 27a that the train loss and test loss improve significantly for $n = 8000$, while the network overfits for smaller $n$. Furthermore, for $n = 2000$, we observed that $\boldsymbol{W}_{10}^\top \boldsymbol{W}_{10}$ exhibits a heavy-tailed ESD with PL_Alpha_Hill$= 1.8$ (Figure 21c), which is less than the $n = 8000$ case of $\approx 1.9$ (see Figure 5c). The key difference in the latter case is that the spike in the ESD is consumed by the bulk, unlike the former where the outlier singular value is almost an order of magnitude away from the bulk.

**Role of regularization constant** ($\lambda$). By considering a sample size of $n = 8000$, and fixing $d = 1000, h = 1500, \eta = 1$ as before, we can observe from Figure 27b that a lower training and test loss is achieved by $\lambda = 10^{-3}$ and $\lambda = 10^{-4}$, but with a large generalization gap (i.e. the difference between train and test loss). Additionally, observe that $\lambda = 10^{-2}$ reasonably balances the test loss and generalization gap. Since the regression procedure does not modify the first layer weights, the choice of $\lambda$ can affect the interpretation of heavy tails leading to good/bad generalization.

**Role of label noise** ($\rho_e$). Figure 27c illustrates a consistent increase in losses for an increase in the additive Gaussian label noise $\rho_e$ from $\{0.1, 0.3, 0.5, 0.7\}$. However, the ESD of $\boldsymbol{W}_{10}^\top \boldsymbol{W}_{10}$ alone does not provide the complete picture to reflect this difficulty in learning. Instead, we observe noticeable differences in the distribution of values (especially outliers) along the diagonal of overlap matrices $\mathcal{O}(\boldsymbol{U}_{W_{10}}, \boldsymbol{U}_{M_9}), \mathcal{O}(\boldsymbol{V}_{W_{10}}, \boldsymbol{V}_{M_9})$ for $\rho_e = 0.3$ (Figure 24c, 24d) and $\rho_e = 0.7$ (Figure 25c, 25d). Especially, the outliers in the former case had smaller values (almost $0.5\times$) than the latter.

**Role of learning rate scheduling.** As a natural extension of selecting a large learning rate at initialization, we analyze the role of employing learning rate schedules (Ge et al., 2019) on the losses and ESD of $\boldsymbol{W}^\top \boldsymbol{W}$. We consider the simple `torch.optim.StepLR` scheduler with varying decay factors ($\gamma$) per step. A smaller $\gamma$ indicates a faster decay in the learning rate $\eta$ per step. We observe from Figure 27d that such fast decays (with $\gamma = 0.2$ and $\gamma = 0.4$) quickly turns a large learning rate to a smaller one and lead to stable loss curves. However, the trends are relatively unstable for $\gamma = 0.6$ and $\gamma = 0.8$ in the early steps.

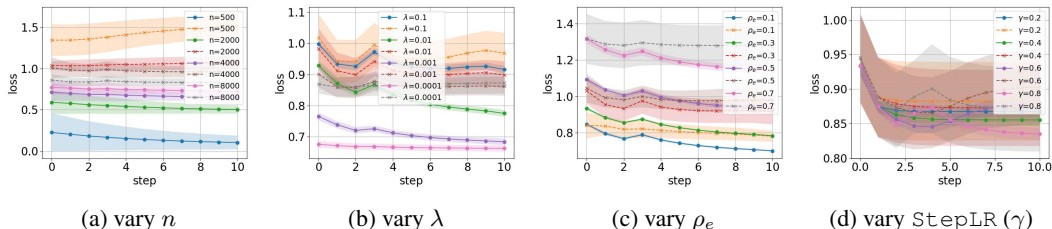

(a) vary $n$      (b) vary $\lambda$      (c) vary $\rho_e$      (d) vary `StepLR` $(\gamma)$

Figure 27: `FB-Adam` train/test loss across 10 steps with varying $n, \lambda, \rho_e$ and `StepLR`$(\gamma)$. (a) Varying dataset size $n$ (b) Varying regularization parameter $\lambda$ for obtaining the second-layer weights, (c) Varying the std.dev ($\rho_e$) of the additive Gaussian label noise, (d) Varying the $\gamma$ parameter which controls the decay of $\eta$. The bold lines indicate train loss and the dashed lines indicate test loss.

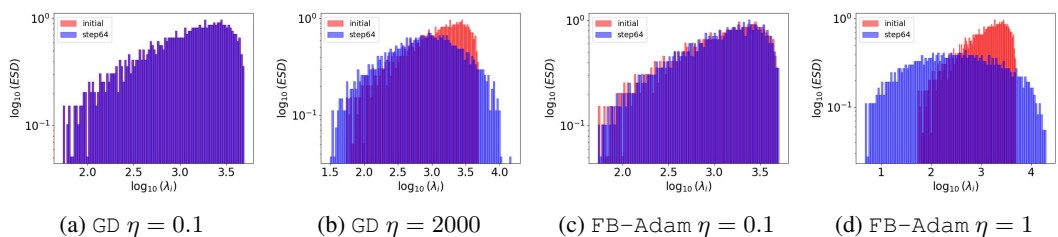

(a) `GD` $\eta = 0.1$      (b) `GD` $\eta = 2000$      (c) `FB-Adam` $\eta = 0.1$      (d) `FB-Adam` $\eta = 1$

Figure 28: Evolution of ESD after 64 steps of `GD`, `FB-Adam` updates with varying $\eta$ and weight normalization. Here $n = 2000, d = 1000, h = 1500, \sigma_* = $ `softplus`$, \sigma = $ `tanh`$, \rho_e = 0.3$.

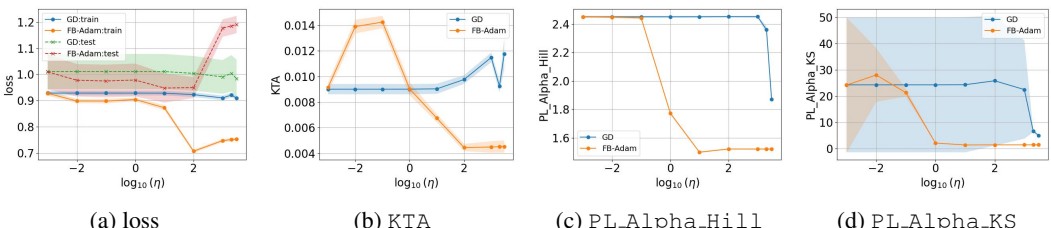

(a) loss      (b) `KTA`      (c) `PL_Alpha_Hill`      (d) `PL_Alpha_KS`

Figure 29: Train/test losses, `KTA`, `PL_Alpha_Hill`, `PL_Alpha_KS` for $f(\cdot)$ trained with 10 steps of `GD`, `FB-Adam` with weight normalization. Here $n = 8000, d = 1000, h = 1500$, $\sigma_* = $ `softplus`$, \sigma = $ `tanh`$, \rho_e = 0.3$.

**Role of weight normalization (WN).** As mentioned in the main text, we employ a weight normalization technique (Huang et al., 2023) after each optimizer update to $\boldsymbol{W}$ as follows: $\boldsymbol{W}_{t+1} = \frac{\sqrt{hd}\boldsymbol{W}'_{t+1}}{\left\| \boldsymbol{W}'_{t+1} \right\|_F}, \boldsymbol{W}'_{t+1} = \boldsymbol{W}_t + \boldsymbol{M}_t$ to ensure that $\left\| \boldsymbol{W}_{t+1} \right\|_F$ is always $\sqrt{hd}$, before the forward pass. Observe from Figure 28 that after 64 steps of `GD`/`FB-Adam` updates with varying $\eta$, the large $\eta$ cases lead to HT ESDs which spread over a wider range of singular values than the non-WN cases (see Table 4). Furthermore, Figure 29 conveys that in the case of `FB-Adam` for $0.01 \leq \eta \leq 100$, the `PL_Alpha_Hill` metric and the mean estimates of `PL_Alpha_KS` tend to exhibit HT ESDs and generalize well. Notice that this range is much broader than the non-WN case in the main text (Figure 5). Additionally, note that the `PL_Alpha_Hill` estimates for $10 \leq \eta \leq 100$ are relatively lower than the non-WN case. Finally, by employing learning rate schedules, we can control the presence of the spike while nudging the bulk of the ESD toward an HT distribution (Figure 30).

### E.5 EFFECT OF USING THE SAME ACTIVATION FUNCTION FOR TEACHER AND STUDENT

In our main paper, we adopt different activation functions for the teacher and student models: $\sigma_* = $ `softplus` for teacher and $\sigma = $ `tanh` for student, Although this distinction introduces some learning challenges for the two-layer NN — making it harder to minimize both training and test loss — it does not affect the key theoretical results of this work, such as the scaling of $\eta$ for a single-step `FB-Adam` update, provided that both activation functions meet the fundamental assump-

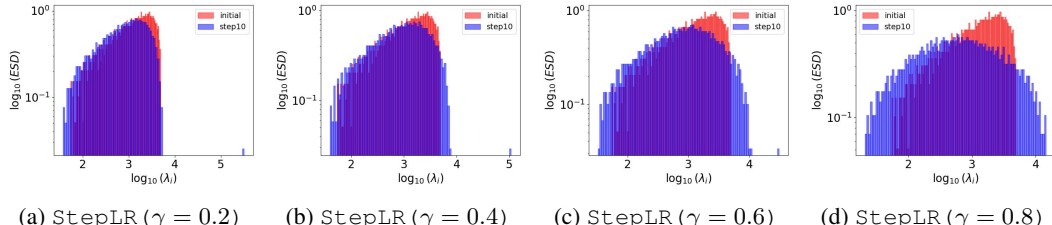

(a) StepLR ($\gamma = 0.2$)    (b) StepLR ($\gamma = 0.4$)    (c) StepLR ($\gamma = 0.6$)    (d) StepLR ($\gamma = 0.8$)

Figure 30: Evolution of ESD of $W^\top W$ after 10 steps of FB-Adam($\eta = 1$) with weight normalization and varying decay rates for StepLR schedule. The decay factor ($\gamma$) is applied after every step. Here $n = 2000, d = 1000, h = 1500, \sigma_* = \texttt{softplus}, \sigma = \texttt{tanh}, \rho_e = 0.3$.

tions. In this section, we use tanh activation for both models to examine whether a more consistent teacher-student setup yields any notable differences. We use the same setup as Section 5.2.

**Correlations between ESD and losses.** Similar to the observations in the main text (see Figure 5) for different activation functions, note that for GD with $\eta \geq 1000$, the KTA increases and PL_Alpha_Hill, mean estimates of PL_Alpha_KS reduces. More importantly, note that the test loss values are relatively smaller (i.e $\approx 0.85$ in Figure 5, and $\approx 0.2$ in this setup). In the case of FB-Adam, we observe a surprising shift in the trend of loss values for $\eta \geq 0.1$. In essence, the range of $\eta$ for which HT ESDs emerge and lead to good generalization has now shrunk to a much smaller range. Especially, the PL_Alpha_Hill value of $\approx 2$ which resulted in good generalization in Figure 5 (relative to losses of the full range of $\eta$), correlates with poorer generalization in this setup. Overall, there seem to be non-trivial dependencies on the choice of activation functions to determine the correlations between HT ESDs and generalization.

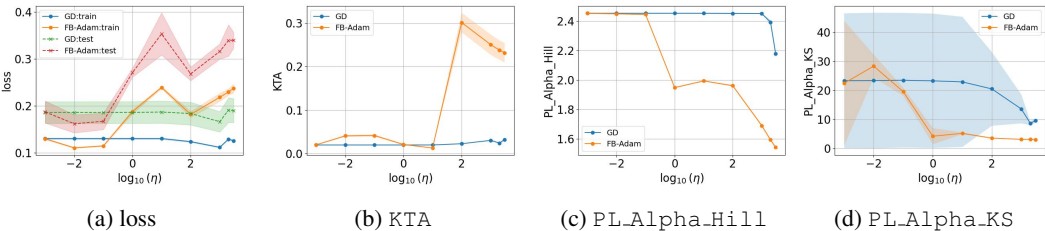

(a) loss    (b) KTA    (c) PL_Alpha_Hill    (d) PL_Alpha_KS

Figure 31: Losses, KTA, PL_Alpha_Hill, PL_Alpha_KS for $f(\cdot)$ trained with 10 steps of GD, FB-Adam, with $n = 8000, d = 1000, h = 1500, \sigma_* = \texttt{tanh}, \sigma = \texttt{tanh}, \rho_e = 0.3, \lambda = 0.01$.

### E.6 APPLICABILITY OF OUR RESULTS ON DEEPER NNS

Beyond two-layer NNs trained on synthetic data, we train `VGG11` with `FB-Adam` on `MNIST` with a constant $\eta = 0.01$ for 10 steps (epochs) to validate our claims. In Figure 32 below, we illustrate that the layers of `VGG11` can exhibit HT ESDs even without stochastic gradient noise during training.

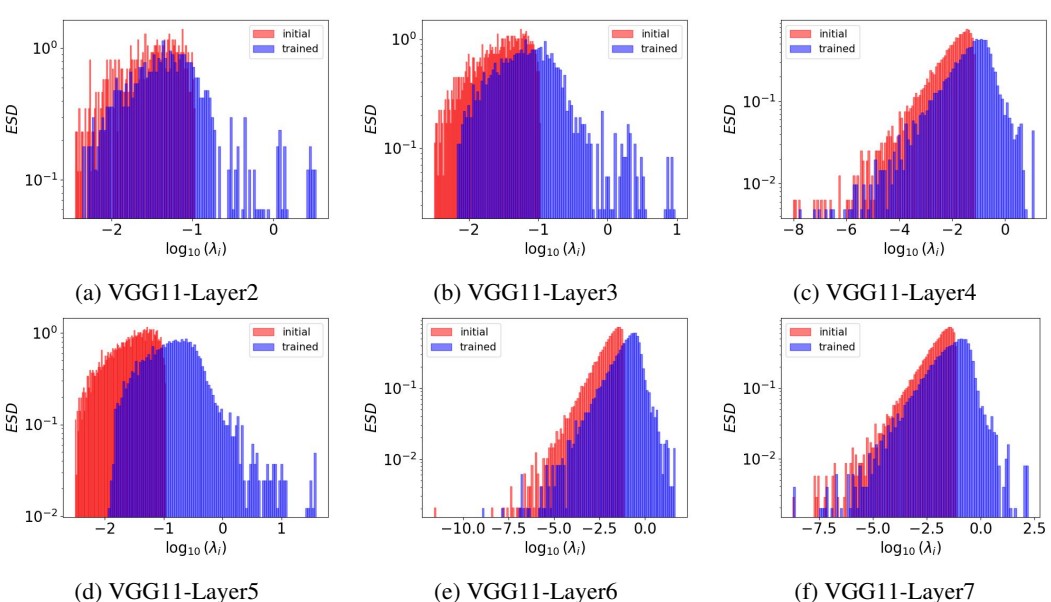

(a) VGG11-Layer2    (b) VGG11-Layer3    (c) VGG11-Layer4

(d) VGG11-Layer5    (e) VGG11-Layer6    (f) VGG11-Layer7

Figure 32: layerwise ESDs of `VGG` after 10 steps with `FB-Adam` and $\eta = 0.01$ on `MNIST`.

### E.7 A NOTE ON THE SUITABLE RANGE OF TAIL INDEX VALUES

Previous works such as Simsekli et al. (2020a); Hodgkinson et al. (2022) studied the limiting distribution of the weight matrix values by modeling the stochastic gradient noise using levy/feller processes. In particular, they showed that smaller tail-index values lead to smaller generalization errors. The underlying idea is that when the weight values tend to an HT distribution, then the resulting ESD is also an HT (see Arous & Guionnet (2008)). Given this explanation, we note that the theoretical results by Simsekli et al. (2020a); Hodgkinson et al. (2022) were shown to hold in practice by their numerical experiments, where the tail index tends to lie in a suitable range (depending on the PL alpha fit approaches) and represents heavy-tailed noise for good generalization. Even though our setting differs significantly from these previous works, the suitability of such a range seems to hold. In particular, when $\eta$ exceeds the thresholds given in Corollary 4.2, the model learns the single index direction and generalizes well by exhibiting HT ESD with a suitable PL Alpha. However, a significant increase in $\eta$ can lead to extreme HTs which are not desirable. The large magnitude of $W$ can also result in large values of MSE on unseen data, especially since the second layer weights are fixed and not selected as per the two-phase training strategy.

## F A NOTE ON ESTIMATORS OF POWER-LAW EXPONENTS

We found that the estimation of power-law (PL) exponents, such as `PL_Alpha_Hill` and `PL_Alpha_KS`, can be sensitive to the scale of singular values in the weight matrix. However, Figure 5c , 5d illustrates a similar trend for both estimates. We believe this does not affect our qualitative interpretation of the relationship between HT, learning rates, and generalization.

Additionally, we highlight a particularly interesting observation. Considering `FB-Adam` based updates with $\eta = 1$ after $t = 10$ steps and after $t = 100$ steps in Table 4, notice that the spike tends to get closer to the bulk in the latter (i.e as training progresses). We have observed a similar behavior for GD (see Appendix E.3) where the spike tends to merge with the bulk. Thus, the estimation of PL exponents should be relatively less affected by such outlier spikes as training progresses.

On a related note, some recent papers have adopted both fitting methods to explore the relationship between HT and generalization and leveraged it to improve model training. For example, Martin & Mahoney (2021a) used `PL_Alpha_KS` to study the relationship between HT and generalization to propose the HT-SR theory, whereas Zhou et al. (2023) proposed a layer-wise $\eta$ selection technique based on layer-wise `PL_Alpha_Hill` estimates of the ESD's during training.

## G  LIMITATIONS AND FUTURE WORK

Currently, our analysis on `FB-Adam` updates does not theoretically analyze the role of large learning rates on the training and test losses after one or more steps. such a rigorous characterization requires new techniques to analyze the regression loss beyond the Gaussian equivalence assumption (Ba et al., 2022). Furthermore, the techniques used to calculate the `PL_Alpha` of ESDs are subject to bias (in the case of the hill estimator) and to relatively larger variance in the case of the KS variant. Similar issues have been discussed in previous works (Yang et al., 2023; Zhou et al., 2023), and analyzing multiple approaches can provide a complete picture of the heaviness of the tails.

To this end, we discuss the following potential future efforts.

**Generalization with `FB-Adam`.**   Recent papers employing a similar setup have analyzed the feature matrix $\overline{Z}$ to rigorously characterize the training and test errors after a one-step GD update (Moniri et al., 2023; Cui et al., 2024). In a similar spirit, the results from our work can be utilized to theoretically explore the spectral properties of $\overline{Z}$ with `FB-Adam`. Additionally, since the spectral properties of $\overline{Z}$ are tightly linked to the ESD of $W_t^\top W_t$, the fundamental question on the necessity of HT ESDs for generalization can be rigorously answered.

**Spectral gap and step complexity.**   In Section 5 we observed that the number of steps required for HT ESD emergence depends on the spectral gap (i.e the distance between the spike and the bulk in our setup) after the first step. While our work establishes the necessity of the spike, further analysis of the relationship between this spectral gap and the step complexity for HT ESD emergence can lead to novel insights.

**HT phenomenon and singular vector overlaps.**   By presenting qualitative results that indicate HT-like distributions along the diagonals of overlap matrices, we aim to bring singular vectors into the picture for future HT phenomenon studies. Particularly, the qualitative differences in the overlap matrices of left and right singular vectors remain to be explored. Furthermore, metrics to quantify the 'spread' of the on/off-diagonal overlap values can present a holistic picture of the interactions between the singular spaces of the weights and optimizer updates.

**Towards analysis with deeper NNs and the '5 + 1' phase model.**   Our work showcased how the '5 + 1' phase model of the ESDs can be studied under simpler two-layer NN settings and reasonably explain the practical NN ESD dynamics. A valuable direction of research is to extend this analysis to deeper NNs (Nichani et al., 2023) with multi-index models as teacher networks, and study the variations in the shapes of ESDs across depth.

**Designing novel techniques to improve generalization.**   Recently, Zhou et al. (2023) employed the layer-wise `PL_Alpha_Hill` metric to design a layer-wise learning rate scheduler based on `SGD`. Based on our observations on the varying effects of $\eta$ for `GD`/`FB-Adam` and the dynamics of ESD evolution, there is immense potential to further improve such schedulers. Furthermore, new regularization techniques to balance the `PL_Alpha_Hill` metrics across layers during training can lead to NNs that satisfy the ESD shape metric criteria to be considered as 'well-trained' by model selection approaches (Martin et al., 2021; Yang et al., 2023). The intriguing consequences of such a technique on the convergence rates and sample complexities can lead to new directions of research.

