# OpenReview forum: "Crafting Heavy-Tails in Weight Matrix Spectrum without Gradient Noise"
_ICLR.cc/2025/Conference — Submitted to ICLR 2025_

### Official Review · Reviewer_piS8 · 2024-10-19

**Soundness:** 3
**Presentation:** 3
**Contribution:** 2
**Rating:** 3
**Confidence:** 4

**Summary:**

The submission studies an interesting problem. When does HT-ESD happen? Why does it lead to good generalization?

The key results are below:

1. Using a toy model,  ADAM after one step can provably induce a spike in spetrum. (thm 4.1)

2. With experiments, the authors show that bulk decay happens (Fig 4), and that spike + decay leads to heavy tail.

3. With experiments, the authors show that successful feature learning happens when the spectrum has the right tail exponent. (fig 5)

##Strength:##

I like the experiment part of this work.  In particular, comparing ADAM vs GD in Fig2 Fig3, and demonstrating the decay in Fig 4.

I think the perspective is novel. The spectrum distribution not only results from the minibatch noise but also from full batch updates.

##Weakness:##

Although I enjoy reading the work, I am not confident that the authors successfully answer the questions asked.

1. The spike after one large update is trivial. Given the backprop structure of neural net, one step update to the weight is rank one. When the step is large, one gets a rank-one spike.

2. The paper demonstrates that decay happens without explaining why. More specifically, does the decay happen just correctly such the spectrum is heavy tailed?

3. The paper demonstrates that the tail index should be in the correct range for the features to be learned. However, no explanation is given. Further, the experiments for this result are synthetic and toy.

I am happy to update my review if the authors could address the above concern.


##Minors:##

What is u in line 241?

**Strengths:**

See summary

**Weaknesses:**

See summary

**Questions:**

See summary

---

> ### Author Response · Authors · 2024-11-17
>
> We thank the reviewer for their feedback and questions
>
> >> The spike after one large update is trivial. Given the backprop structure of neural net, one step update to the weight is rank one. When the step is large, one gets a rank-one spike.
>
> - We humbly note that the gradient matrix $G_0$ of the weight $W_{t=0}$ (i.e. the first step full-batch gradient) is not exactly rank-1, but approximately rank-1. The proof of this result requires careful analysis and can be non-trivial (see proposition 2 of Ba. et.al). In particular, $G_0$ can be approximated by a rank-1 matrix under the spectral norm for sufficiently large n, d. These results for one-step Gradient Descent are then leveraged to analyze the scale of learning rates for the bulk + spike spectrum in $W_{t=1}$.
>  - In this paper, one of our main contributions is an analysis for Adam, where the optimizer update is not the full-batch gradient. More importantly, we formulate a sign-descent approximation for the Adam update and present an approach to compute its spectral and Frobenius norms. Next, we focus on the scale of the learning rate, which requires the results on the norms (from the previous analysis). Especially, these norms tend to differ for GD/Adam due to the sign-like update in Adam. Our results show that learning rates of constant order are sufficient to obtain the bulk+spike with Adam. This result is novel to the literature and clearly shows why Adam requires smaller values of learning rates when compared to GD for resulting in such a bulk+spike ESD (see also Corollary 4.2).
>
> >> The paper demonstrates that decay happens without explaining why. More specifically, does the decay happen just correctly such the spectrum is heavy tailed?
>
> - This is an interesting question. Based on our experiments, we have observed that: the relative position of the rank-one spike from the bulk determines the rate of emergence of HT ESDs. Formally, observe from the full Table 4 in Appendix D.2, that $\eta=0.1$ requires $10,000$ steps for the ESD to transition into an HT whereas $\eta=1$ requires just $10$ steps. This indicates that when the rank-one spike after the first step is sufficiently far from the bulk, then HT ESDs can emerge early in training. We believe that our analysis of the singular vector alignments (Figure 6) presents some initial intuitive understanding of this decay. Intuitively, consider the first step update which results in a rank-one spike in the weight matrix. The singular vector of $W_{t=1}$, which corresponds to this spike tends to align with that of the update matrix $M_0$ (notation as per Eq 12). Note that this spike in $W_{t=1}$ will affect the gradient matrix spectrum of the second step and so on. Through these interactions between the weight matrix spectrum and optimizer update matrix spectrum, the bulk of $W_t$ decays and results in an HT ESD.
> - Theoretically characterizing the role of this evolving spectral gap during training and the onset of HT is an important future direction of research. Our work focuses on presenting extensive preliminary experiments to validate this behavior and motivate future efforts to build on top of it.
>
> >> The paper demonstrates that the tail index should be in the correct range for the features to be learned. However, no explanation is given.
>
> - **Regarding the range of values for tail index:** Thanks for pointing this out. An explanation for such a suitable range of tail-index values was shown by Simsekli et al [2] (and similar works such as [3]). The main idea in these approaches is to model the stochastic gradient noise using levy processes and show that smaller tail-index values lead to smaller generalization errors. Please note that these settings do not explicitly study the ESD of the weight matrix, but study the limiting distribution of the weight matrix values. The underlying idea is that when the weight values tend to an HT distribution, then the resulting ESD is also an HT (see [4]). Given this explanation, we note that the theoretical results by [2,3] were shown to hold in practice by their numerical experiments, where the tail index tends to lie in a suitable range (depending on the PL alpha fit approaches) and represents heavy-tailed noise for good generalization.
>  -  Even though our setting differs significantly from these previous works, the suitability of such a range seems to hold. In particular, when $\eta$ exceeds the thresholds given in Corollary 4.2, the model learns the single index direction and generalizes well by exhibiting HT ESD with a suitable PL Alpha. However, increasing $\eta$ to extremely large values can lead to extreme HT's which are not desirable. The large magnitude of the hidden layer weight matrix in this case can also result in large values of MSE on unseen data, especially since the second layer weights are fixed and not selected as per the two-phase training strategy.
>
> **We have added this discussion (to Appendix E.7) in the revision.**

---

> ### Author Response · Authors · 2024-11-17
>
> >> Further, the experiments for this result are synthetic and toy.
>
> - **Regarding the synthetic setup:** We would like to kindly emphasize that our study aims to systematically analyze the widely known HT ESD phenomenon in practical networks. In essence, our setup aims to analyze and reason about the already known correlations between HT ESDs and generalization, albeit in a simpler setting. As pointed out by the reviewer, our work indeed demonstrates that the tail index should be in the correct range for good generalization. More importantly, such an observation aligns with the empirical results of Martin et al [5], who showed that extremely heavy-tailed ESD does not lead to good generalization and that a suitable range of PL Alpha is desirable.
>
> - We also note that such a setup allows us to rigorously characterize the learning rates for the emergence of the spike after one step Adam (as per Theorem 4.1, Corollary 4.2). Thus, laying the theoretical foundation for this bulk+spike phase allows us to confidently analyze the dynamics of ESD evolution towards HT.
>
> - In addition to the applicability, we highlight that our results clarify that gradient noise is not necessary for the emergence of HT ESDs. This result was not previously known/formally studied. More importantly, previous works have largely assumed that stochastic gradient noise is necessary for HT emergence. Our results clarify this prevailing notion for the first time. Furthermore, we also presented evidence in Appendix E.6 that practical networks such as VGG can indeed exhibit HT ESDs with large enough learning rates and without gradient noise. Thus showcasing the applicability of insights beyond 2 layer networks.
>
> >> What is u in line 241?
>
> In line 241, $u_1$ is the first principal component of $W$ (i.e. the singular vector corresponding to the largest singular value).
>
> **References:**
>
> [1] Ba, Jimmy, et al. "High-dimensional asymptotics of feature learning: How one gradient step improves the representation." Advances in Neural Information Processing Systems 35 (2022): 37932-37946.
>
> [2] Simsekli, Umut, et al. "Hausdorff dimension, heavy tails, and generalization in neural networks." Advances in Neural Information Processing Systems 33 (2020): 5138-5151.
>
> [3] Hodgkinson, Liam, et al. "Generalization bounds using lower tail exponents in stochastic optimizers." International Conference on Machine Learning. PMLR, 2022.
>
> [4] Gerard Ben Arous and Alice Guionnet. The spectrum of heavy tailed random matrices. ´ Communications in Mathematical Physics, 278(3):715–751, 2008.
>
> [5] Martin, Charles H., and Michael W. Mahoney. "Implicit self-regularization in deep neural networks: Evidence from random matrix theory and implications for learning." Journal of Machine Learning Research 22.165 (2021): 1-73.

---

> > ### Comment · Reviewer_piS8 · 2024-11-22
> > **Response to the authors**
> >
> > I thank the authors for the response.
> >
> > -“ our main contributions is an analysis for Adam”
> >
> > Up to my knowledge, any nonlinear operation (such as relu, or sign as in ADAM) can be smoothed out at initialization, and hence the rank is essentially determined by xa^T, where x is the input and a is the activation.
> >
> > -"Intuitively, consider the first step update which results in a rank-one spike in the weight matrix. The singular vector of
> > , which corresponds to this spike tends to align with that of the update matrix  (notation as per Eq 12). "
> >
> > I see the point by the authors that the spike effect somehow lasts more than the first iteration. Still, it is unclear whether the decay or the spike is stronger, and why the shape would be HT.
> >
> > -"levy processes and show that smaller tail-index values lead to smaller generalization errors"
> >
> > I think this is exactly the case the result in this paper should differ from. Specifically, [2] is studying SGD noise, which is different from the spectral being heavy tailed, and this difference is the main contribution of this work.
> >
> >
> > I thank the reviewer for the response and I like the topic in this submission; however, I didn't get much information / insight after reading the work or the response.

---

> > > ### Author Response · Authors · 2024-11-22
> > > **Response to the reviewer**
> > >
> > > **We thank the reviewer for engaging in a discussion. We would like to clarify some misunderstandings below.**
> > >
> > > >> Up to my knowledge, any nonlinear operation (such as relu, or sign as in ADAM) can be smoothed out at initialization, and hence the rank is essentially determined by $xa^T$, where x is the input and a is the activation
> > >
> > > **Answer:** Please note that (as per the papers notation) the gradient approximation is given by: $ay^\top X$, where $a$ is the final layer weight, $y$ is the target label, and $X$ is the input data (see Appendix B.1, equation (22)). Could the reviewer please explain how they obtained their formulation in their response?
> > >
> > > - Furthermore, it is not straightforward to formally show the effect of the sign operation on the full-batch gradient to obtain the Adam update. Especially to calculate the norms of the Adam update, which is required to study the critical learning rates for the Bulk+spike phase. To the best of our knowledge, we are not aware of any formal proofs for such analysis (as also acknowledged by other reviewers). We also humbly request the reviewer to provide references to such non-linear operations on the gradient matrices in case we might have missed any.
> > >
> > > >> I see the point by the authors that the spike effect somehow lasts more than the first iteration. Still, it is unclear whether the decay or the spike is stronger, and why the shape would be HT
> > >
> > > **Answer:** Formally, our results showcases the HT-like decay of overlap values between the singular bases (Section 5.3) of $W$ and optimizer updates, which tend to correlate with the emergence of HT ESDs in $W$. Intuitively, one can think of the "energy" (pertaining to the singular value/spike) of the singular vector of the Adam update to be spread/diffused into the singular bases of the weight matrix $W$. Thus, resulting in a HT-like decay. We hope this clarifies the confusion.
> > >
> > > We also kindly emphasize that the motivation of our work/setting is to be able to facilitate such multi-step spike analysis in the first place. Please note that theoretically analyzing the dynamics of ESD shape evolution toward HT is still an open-question that needs to be formally proved. We discussed this in Appendix G in more detail. In this context, our work takes a step towards understanding the HT-ESD shape via a qualitative analysis of the singular bases alignments.
> > >
> > > >> I think this is exactly the case the result in this paper should differ from. Specifically, [2] is studying SGD noise, which is different from the spectral being heavy tailed, and this difference is the main contribution of this work.
> > >
> > > **Answer:** We would like to clear a misunderstanding. Kindly note that [2] studies the HT nature of SGD noise, which in turn translates into the HT-nature of the weight values. Furthermore, based on the formal results in [4], when the weight values (i.e iterates) tend towards a HT distribution then the ESD also tends towards a HT distribution. Thus, our work is connected to these previous results at a fundamental level where HT nature of iterates and HT-ESD's (both) correlate with generalization.
> > >
> > > Furthermore, our paper does not claim to disprove the previous results. Instead, our work aims to study the HT ESDs by instead focusing on the early phases of training and without any gradient noise (line 48-50 in main paper) (**which was previously not known in the literature**). In summary, it is an alternative perspective to understand the HT-ESD emergence. We hope this clarifies the confusion.

---

> > > > ### Author Response · Authors · 2024-11-24
> > > > **Requesting feedback on the clarifications**
> > > >
> > > > Gentle ping to request feedback on our response and clarification. We are happy to answer any further questions.
> > > >
> > > > - Authors

---

> > > > ### Author Response · Authors · 2024-11-30
> > > > **Requesting feedback on the clarifications**
> > > >
> > > > Dear Reviewer piS8,
> > > >
> > > > As the discussion deadline is approaching, we kindly request feedback on our response and clarifications. We are happy to answer any further questions.
> > > >
> > > > Thanks again for your time.

---

### Official Review · Reviewer_xTm5 · 2024-11-04

**Soundness:** 3
**Presentation:** 4
**Contribution:** 2
**Rating:** 6
**Confidence:** 4

**Summary:**

This paper is interested in understanding the evolution of the empirical spectral distribution for the first-layer weight matrix in a two-layer neural network during training. The paper considers a student teacher (single index model) with Gaussian. It shows that early-stage feature learning occurs for a much smaller step size for Adam compared to GD.

The paper also empirically studies the spectrum's evolution during further training and claims the existence of the initial spike is needed to transition to the heavy-tailed phase. Finally, the paper looks at various kernel alignment metrics, their evolution, and correlation with generalization.

**Justification for Score**

I think overall the paper presents intersting insights for a different optimizer. Hence I think it is worth accepting. However, the paper doesn't always position itselves in the most informative way compared to prior work and I have concerns about the significance of the spike that has been seen. Hence I do not give it a higher score.

**Strengths:**

**Novelty:** As far as I know, the theoretical works on the early-stage emergence of spikes have been primarily limited to gradient descent. No theoretical work I know of analyzes Adam. Hence I think this is a novel contribution of the paper. Additionally, the insight that Adam requires a smaller step size than gradient descent to see the spike is important.


**Clarity:** The paper is mostly well-written. My one concern is that when people talk about the spiked structure (Ba et al, Moniri et al). they mean for the features $F = \sigma(WX)$ and not for the weight matrix $W$. This was confusing at first.

**Weaknesses:**

**Originality:** In terms of originality, the paper should do a better job of positioning itself with respect to Martin and Mahoney,  Ba et al., and Moniri et al. I think this is important.


**Significance:** I think the result on smaller step size results in spikes is significant. If this were a result as presented in Moniri et al. However, as a result is currently presented, I am not sure the theory result says enough. Specifically,

1. The paragraph after the corollary is clear to me. The step size scales are different for the spectral norm and the Frobenius norm. How do we rationalize this to show that there is a spike?

2. Spikes in $W$ do not necessarily correspond to spikes in $F$. Especially if you have Gaussian Data. See [1] (flipping the role of $W$ and $X$).

[1] Wang, Zhichao, Denny Wu, and Zhou Fan. "Nonlinear spiked covariance matrices and signal propagation in deep neural networks." arXiv preprint arXiv:2402.10127 COLT (2024).

**Questions:**

1. Generalization with Heavy-Tailed models. Do the authors know of any works that theoretically characterize the generalization error of models with heavy tails? Even in the regression setting I know of [2]. Also, do the authors know about papers that consider generalization error for models with spiked covariance (besides the Ba et al. 2023 paper), again including regression.

[2] Wang, Yutong, Rishi Sonthalia, and Wei Hu. "Near-interpolators: Rapid norm growth and the trade-off between interpolation and generalization." International Conference on Artificial Intelligence and Statistics. PMLR, 2024.

2. Emergence of heavy tails. For $\eta=0.1$, the experiment needed $t \sim 10^4$. However, the authors claim $t \sim 10^4$ was not enough for $\eta = 0.01$. I imagine we need more steps, maybe even something like $t \sim 10^7$.

---

> ### Author Response · Authors · 2024-11-17
>
> We thank the reviewer for their constructive feedback.
>
> >>  In terms of originality, the paper should do a better job of positioning itself with respect to Martin and Mahoney, Ba et al., and Moniri et al. I think this is important.
>
> - Thank you for the suggestion. The essence of Martin and Mahoney's work lies in their empirical '5+1' phase transition model of the weight matrix ESDs during training. Thus, allowing them to study the role of ESD evolution and correlations between HT and good generalization. We clarify that (as also mentioned in the introduction), the motivation of our work is to systematically study such ESD evolution into HT and its relationship with generalization in 2-layer NNs.
>  - Based on this 5+1 phase model, the bulk + spikes phase of the ESD is the precursor to a decay into HT. Therefore, our work first studies the setting under which the ESD can transition from a random-like (Marchenko–Pastur) distribution at initialization to a bulk + spike. The work of Ba et al. is extremely relevant in this context since they theoretically derived the critical learning rate for resulting in a bulk+spike ESD after the first step of GD. Furthermore, they show how this spike aligns with the target direction of the single-index teacher model. Thus, leading to feature learning in a two-layer NN. We emphasize that this relationship between the bulk+spike ESD (in the context of HT emergence) and feature learning, is where we connect these two lines of work.
> - Moniri et al.'s work primarily focuses on the spectrum of the feature covariance matrix, unlike our work which focuses on the ESD of weights. More importantly, we do not analyze feature covariance matrices and their spectra to derive precise asymptotic prediction errors as Moniri et al, but instead focus on correlations between HT ESDs (the tail indices and learning rates) and generalization via extensive empirical analysis in our setup. We discuss these aspects in Appendix G.
>
> In summary, based on the relevant motivation to study the 5+1 phase model by Martin and Mahoney, we extend Ba et.al's results on the critical learning rate to the case of one-step FB-Adam update (which aligns closely with practical training strategies). Next, we show how such bulk + spike ESD facilitates the emergence of HT ESD early in the training and analyze its correlations with generalization.
>
> >> The step size scales are different for the spectral norm and the Frobenius norm. How do we rationalize this to show that there is a spike?
>
> As per equation (12) in our paper, one can (generically) represent the first step update as: $W_1 = W_0 + M_0$. Here $M_0 = -\eta\widetilde{G_0}$ is the first-step Adam optimizer update which can be considered as the signal matrix, whereas $W_0$ is a random normal matrix. The study of the occurrence of a spike in the ESD of $W_1$ is a classic problem in the spiked matrix model literature [1]. From Corollary 4.2 in the main paper, we know that when $ \eta = \Theta(1) $, then we have $ ||-\eta \widetilde{G_0} ||_F = || W_1 - W_0 ||_F \asymp  || W_0 ||_F = \Theta(\sqrt{hd}) $. Intuitively, observe that when $ || -\eta \widetilde{G_0}||_F \asymp ||W_0||_F $, the singular value (corresponding to the rank-1 singular vector) of the update matrix has a magnitude that is of the same order as $||W_0||_F$. This large "strength" of the signal facilitates the occurrence of the spike. Finally, we would also like to refer to the classical results on the BBP phase transition [14] and its follow-up works to characterize the strength of the low-rank signals for such spikes to appear.
>
> >> Spikes in $W$ do not necessarily correspond to spikes in $F$. Especially if you have Gaussian Data
>
> We acknowledge that spikes in the weight matrix $W$ do not necessarily correspond to spikes in features $F$. These are two different spiked models: the spiked weight model (which is the focus of this work) and the spiked feature covariance model (such as layer-wise conjugate kernels). Their spectral properties and interconnections are discussed in [2][3], as well as in Appendix A of [4]. Given this clarification, we underscore that our results to characterize the learning rate scales for Adam are relevant for the bulk+spike weight matrix ESD. The effects of such Adam updates on the feature covariance matrix, (in particular the spikes in the ESD), and generalization requires rigorous independent analysis. We present an initial discussion on this line of work in Appendix G.

---

> ### Author Response · Authors · 2024-11-17
>
> >> Generalization with Heavy-Tailed models. Do the authors know of any works that theoretically characterize the generalization error of models with heavy tails? Also, do the authors know about papers that consider generalization error for models with spiked covariance (besides the Ba et al. 2023 paper)
>
> Based on the clarification (in the above response), we discuss papers related to generalization with Heavy-Tailed models by dividing them into spike weight models and spiked feature covariance models.
>
> - (spiked weights): The works of [5][6] attribute the HT-ESD in weight matrices during training to HT sgd noise and proposed generalization bounds which depend on the tail-index of the limiting HT distributions. These papers leverage stochastic differential equations (SDE) based on levy/feller processes and fractal geometry to derive the bounds. The work of [7] studies the algorithmic stability of the heavy-tailed SDE-based optimization process via Wasserstein stability bounds which are then used to derive generalization bounds. On a related empirical note: The work of [8] treats the HT ESD as an extreme case of bulk+spike and analyzes the subspaces of singular vectors that constitute the spikes. In particular, they show the relationship between such subspaces and the hidden directions of the multi-index models to analyze the generalization performance of NNs.
>
>  - (spiked features): The works of [9, 10, 11] theoretically and empirically explore the generalization and robustness with heavy tails in spiked feature covariance models in self-supervised learning and adversarial learning. Work in [12] theoretically explores the benign overfitting phenomenon in linear regression and builds a relationship between the test error with heavy tails in spiked feature covariance. On a related note, similar to the work of Moniri et.al (as mentioned by the reviewer), the work of [13] analyzes generalization with spiked feature spectra and its benefits over the random features model.
>
> >>  Emergence of heavy tails. For $\eta=0.1$ , the experiment needed $t\sim 10^4$ . However, the authors claim  was not enough for $\eta=0.01$. I imagine we need more steps, maybe even something like  $t\sim 10^7$
>
> Thank you for pointing this out. Based on the request of the reviewer, we were able to run an experiment with FB-Adam and $\eta=0.01$ for $t=10^6$ steps (experiment ran for > 24 hrs) with the same setup as per the main paper and observed that an HT ESD can indeed emerge after such infinitely long training. Based on this observation we note the following:
>
> - In our setup, $h=1500$, and the value of $\eta=0.01$ tends to lie in the range of $(\Theta(1/\sqrt(h)), \Theta(1))$ (as per corollary 4.2), considering the role of scaling constants for $\Theta$, as per the setting. We believe that this choice of $\eta=0.01$ leads to a scenario where a spike does not emerge after one step but the magnitude of gradient updates over infinite training steps i.e., $t=10^6$ can lead to a bulk transitioning to HT ESD. Given this observation, extremely small $\eta$ might not be able to exhibit such changes to the ESD even after $t > 10^6$ steps. See also Corollary 5.3 [8] for a related formal result.
>  - More importantly, our claim and focus on how the spike leads to HT ESD after "finite" subsequent steps stays the same and is not affected by this observation.
>  - To avoid confusion, we will modify lines 104, 262, and 365 (first paragraph in Section 5.1) to remove the explicit statements on the "necessity of spike after the first update" for HT ESDs and instead focus on how such a spike can facilitate the emergence of HT. We will also explicitly state that for learning rates that might lie in $(\Theta(1/\sqrt(h)), \Theta(1))$ (as per corollary 4.2), the ESD might not exhibit a spike after one step but can potentially lead to HT ESD after infinitely long training steps.
> - Overall, our main message on how the presence of a spike (i.e. the bulk + spike ESD) after the first step facilitates the transition to HT in the early phases of training remains the same (as also mentioned in the abstract). Thanks again for bringing this point up and helping us avoid such confusion.
>
> **Update: The revised version has been uploaded.**
>
> **References:**
>
> [1] Loubaton, Philippe, and Pascal Vallet. "Almost sure localization of the eigenvalues in a Gaussian information plus noise model. Application to the spiked models." (2011)
>
> [2] Wang, Zhichao, Denny Wu, and Zhou Fan. "Nonlinear spiked covariance matrices and signal propagation in deep neural networks." arXiv:2402.10127 (2024).
>
> [3] Dandi, Yatin, et al. "A Random Matrix Theory Perspective on the Spectrum of Learned Features and Asymptotic Generalization Capabilities."arXiv:2410.18938 (2024).
>
> [4] Zhou, Yefan, et al. "Temperature balancing, layer-wise weight analysis, and neural network training." NeurIPS 2023
>
> [5] Simsekli, Umut, et al. "Hausdorff dimension, heavy tails, and generalization in neural networks." NeurIPS (2020)

---

> > ### Author Response · Authors · 2024-11-17
> >
> > [6] Hodgkinson, Liam, et al. "Generalization bounds using lower tail exponents in stochastic optimizers." ICML, 2022.
> >
> > [7] Raj, Anant, et al. "Algorithmic stability of heavy-tailed sgd with general loss functions." ICML, 2023.
> >
> > [8] Wang, Zhichao, et al. "Spectral evolution and invariance in linear-width neural networks." NeurIPS 2023
> >
> > [9] Nassar, Josue, et al. "On 1/n neural representation and robustness." NeurIPS 2020
> >
> > [10] Agrawal, Kumar K., et al. "$\alpha $-ReQ: Assessing Representation Quality in Self-Supervised Learning by measuring eigenspectrum decay." NeurIPS 2022
> >
> > [11] He B, Ozay M. Exploring the gap between collapsed and whitened features in self-supervised learning, ICML 2022
> >
> > [12] Bartlett, Peter L., et al. "Benign overfitting in linear regression." PNAS 117.48 (2020): 30063-30070.
> >
> >
> > [13] Cui, Hugo, et al. "Asymptotics of feature learning in two-layer networks after one gradient-step." ICML, 2024
> >
> > [14] Baik, Jinho, et.al "Phase transition of the largest eigenvalue for nonnull complex sample covariance matrices." (2005): 1643-1697.

---

> > > ### Comment · Reviewer_xTm5 · 2024-11-24
> > >
> > > Thank you for the responses and clarifying my questions.
> > >
> > > The references to heavy-tailed (and the longer-run experiment) are interesting.
> > >
> > > I think this paper has some good ideas and could be worth accepting. Hence I will keep my score.

---

> > > > ### Author Response · Authors · 2024-12-04
> > > >
> > > > Thank you for supporting our work.

---

### Official Review · Reviewer_bo67 · 2024-11-04

**Soundness:** 4
**Presentation:** 3
**Contribution:** 3
**Rating:** 8
**Confidence:** 4

**Summary:**

The paper aims to investigate the relationship between heavy tails in the weight spectrum distribution of neural networks and the ability of the network to generalize over unseen samples. The authors set up a teacher-student setting where a two-layer neural network learns a single-index model. First, the paper shows that for a sufficiently large step size, the full-batch Adam can align with the teacher's direction in just one step, corresponding to a spike in the ESD emerging from the initial mass. The analysis continues by empirically showing that the dynamics of the ESD evolves from the bulk to a heavy tail distribution. Finally, the paper shows the connection between the ESD heavy tails and the orientation of the singular vector of weight updates.

**Strengths:**

- To my knowledge, this paper is the first to study the HT-generalization correspondence in the single-index teacher-student setting. Altough not bringing any revolutionizing idea, the paper establishes an important first step towards understating this phenomenon within the community.
- The analysis is comprehensive and precise, also taking into account techniques used in practice such as weight normalization and learning rate schedulers.

**Weaknesses:**

- The paper is showing mostly empirical result. In this theoretical setting one might expect to have more theoretical support of the claims.

**Questions:**

- In line 241, shouldn't we divide the similarity by the norm of $u_1$? Otherwise it can grow even if $u_1$ and $\beta$ are not getting more aligned
- In line 308 you talk about a sweet for $\eta$, but I can't see it in the figure, can you clarify?

Minor:
- Line 297: $\eta=0.1$, but the figures says $\eta=1.$

---

> ### Author Response · Authors · 2024-11-16
>
> We thank the reviewer for their encouraging feedback.
>
> >> (a) The paper is showing mostly empirical result. In this theoretical setting one might expect to have more theoretical support of the claims.
>
> - We agree that our theoretical results are currently applicable to the bulk+spike ESD after one step of Adam. Nonetheless, we hope that this extension to Adam is of broader interest to the community as it potentially unlocks new approaches to studying adaptive gradient methods. Additionally, we discuss the limitation of extending our analysis to multiple steps in Appendix G. In particular, since we use large learning rates, our setup is not amenable to continuous time approximation techniques to potentially study the ESD evolution. Furthermore, the Gaussian equivalence of the weight matrices does not hold in this regime which allowed Ba et.al [1] to study the role of finite training steps on the weight matrix and generalization. To this end, we kindly note that establishing theoretical results to rigorously characterize the dynamics of the ESD evolution requires an independent analysis of its own (especially in this large lr regime).
>
>
> >> (b) In line 241, shouldn't we divide the similarity by the norm of $u_1$ ? Otherwise it can grow even if $u_1$ and $\beta$ are not getting more aligned?
>
> - Since $u_1$ is the singular vector corresponding to the top singular value (i.e a column of an orthonormal matrix), we consider it to have a unit norm.
>
>
> >> (c) In line 308 you talk about a sweet for $\eta$, but I can't see it in the figure, can you clarify?
>
> - Thanks for bringing this up. We can consider the following cases to better understand the plot for Adam in Figure 3. Case (1) For $\eta < 0.01$, generalization does not improve and we do not see an increase in KTA/alignment. Case (2) For $0.01 \le \eta \le 1$, generalization starts to improve and KTA/alignment values increase. Case (3) For $\eta > 1$, generalization degrades and there is a slight reversal in the trend for KTA. Note that due to the two-phase training strategy, the large magnitude of the first step update in Case (3) can lead to lower training loss (since the last layer is computed based on a closed-form solution Eq(6) and the paragraph below it). However, during test time, such large magnitudes of values in the hidden layer weights seem to result in higher test MSE loss. We have added these points to the revised article to improve clarity.
>
> **Minor comments on $\eta$:** We have fixed the $\eta=1$ value in the revision. Thanks for pointing it out.
>
> **References:**
>
> [1] Ba, Jimmy, et al. "High-dimensional asymptotics of feature learning: How one gradient step improves the representation." Advances in Neural Information Processing Systems 35 (2022): 37932-37946.

---

> > ### Comment · Reviewer_bo67 · 2024-12-02
> >
> > I confirm my score and recommend the acceptance.

---

> > > ### Author Response · Authors · 2024-12-04
> > >
> > > Thank you for supporting our work.

---

### Official Review · Reviewer_uys5 · 2024-11-05

**Soundness:** 2
**Presentation:** 2
**Contribution:** 2
**Rating:** 3
**Confidence:** 4

**Summary:**

The paper studies the heavy tail phenomenon of the spectrum of weight matrices and its impact on generalization.  The paper presents a theorem for single step of Adam which shows an emergence of "Bulk + spike" spectrum. It futher empirical evaluates how this spectrum evolves into a heavy tail.

**Strengths:**

a) The paper extends the single step analysis of GD Ba et. al. for Adam.
b) The observation that bulk + spike evolves into a heavy tail is interesting.

**Weaknesses:**

a) The paper lacks sufficient motivation for the problem setting and the relevance of the chosen algorithm, leaving it unclear why this is an important quantity to study. Notably, both the heavy-tail mechanism and strong generalization stem from learning the single-index direction, making it uncertain whether the heavy tail is simply a byproduct of good generalization or its cause. Consequently, it is challenging to assess whether this problem setting and analysis truly captures the influence of the heavy-tail spectrum on generalization.

b) The empirical analysis is restricted to single-index teachers and two-layer networks, which limits the study's scope and makes it insufficient to determine whether any findings are broadly applicable.

c) The theoritical analysis does not capture how "BULK + spike" transition into a heavy tail phenomenon. Hence the paper falls short of theoritically understanding the emergence of HT-ESD - the theoritical analysis is only captures the BULK + spike shape of spectrum after a single step of Adam.

Minor :
- Make a consistent notation of $\beta^{*}$ through out the paper.
- The inline math can be formatted better to ensure better readbility.

**Questions:**

discussed above

---

> ### Author Response · Authors · 2024-11-16
>
> We thank the reviewer for their feedback on our work.
>
> >> a) The paper lacks sufficient motivation for the problem setting and the relevance of the chosen algorithm, leaving it unclear why this is an important quantity to study...
>
> **Regarding the motivation and problem setting:** Our work is based on the premise that there exists a **correlation** between HT-ESD and generalization in deep NNs (ex: the implicit self-regularization work by Martin and Mahoney [1] and also Simsekli et.al [2]). Our motivation for the problem setting and the chosen algorithm is based on the relevant work by Wang et.al [3]. In particular, they presented preliminary empirical results on training 2-layer NNs to learn multi-index models and the relationships between HT ESD and good generalization. They showed that mini-batch training with Adam can facilitate the emergence of HT ESD but no particular theoretical result for such emergence was given. Based on these observations and previous research, our setting is well-motivated to study 2-layer NNs and single-index models to explicitly characterize the role of optimizers and learning rates for such HT ESD emergence.
>
> **Regarding whether the HT is simply a byproduct of good generalization or its cause:** We agree with the reviewer that the ESD needs to capture the directions of the single-index model. In the more generic setting of multi-index models, the preliminary experimental results of Wang et.al [3] concluded that HT ESDs are not sufficient for good generalization. Our results present a more nuanced perspective on this topic. We observe from Figure 5 that when $\eta=1$, the PL Alpha Hill value after $10$ steps indicates an HT ESD and a clear reduction in test loss can be observed (compared to smaller $\eta$ which does not result in HT ESD). Furthermore, Figure 24 shows that the test loss gradually decreases during these 10 steps. Next, considering the setup in Section 5.2, the causality/by-product aspect of HT ESD can be understood by the following breakdown: (1) For $\eta < 0.01$, generalization does not improve and HT ESD does not occur. (2) For $0.01 \le \eta \le 1$, generalization improves and HT ESD occurs. (3) For $\eta > 1$, generalization degrades and HT ESD still occurs. Case (3) is interesting because the spike after the first step facilitates a decay into HT ESD, whereas the extremely large scale of updates to the weight matrix hinders generalization. These observations clarify that: **(a) HT ESDs are not a byproduct of generalization, but (b) tend to be necessary for a suitable range of learning rate values.** From a practical perspective, these observations align with the results of Martin and Mahoney [1] where they show that truncated PL fit values in the range of $\in [2,4]$, relate to HT ESD and good generalization.
>
> **Regarding the setting and analysis to truly capture the influence of the HT ESD on generalization:** By choosing a setup with two-layer NN and single-index models, our work (1) allows the study of HT ESDs and generalization without gradient noise, (2) analyzes the role of learning rates, (3) analyzes the role of techniques such as weight-normalization and learning rate schedules. More importantly, as per our response above, our setup is rich enough to study the correlations between HT-ESD and generalization that were observed with deep networks in practical settings. Additionally, our setting also allows us to study how a single spike can transition the bulk into an HT ESD. Thus allowing a systematic study of the evolution phases of the spectrum (or more formally, the 5+1 phases of spectrum evolution by Martin and Mahoney [1]).
>
> >>  (b) The empirical analysis is restricted to single-index teachers and two-layer networks, which limits the study's scope and makes it insufficient to determine whether any findings are broadly applicable.
>
> As per the discussion above, the motivation of our work is to study and reason about the already known correlations between HT ESDs and generalization in deep NNs, albeit in a simpler setting.
>
> Our setup allows us to rigorously characterize  $\eta$ for the emergence of the spike after one step Adam (as per Theorem 4.1, Corollary 4.2) and lay the theoretical foundation for this bulk+spike phase.  Thus allowing us to confidently analyze the dynamics of ESD evolution towards HT. Furthermore, these results extend the GD-based results of Ba et.al [4] to the case of Adam. Thus, making it of interest to a broader audience as we move closer to practical settings.
>
> Our results also clarify that gradient noise is not necessary for the emergence of HT ESDs. More importantly, since previous works have largely assumed that stochastic gradient noise is necessary for HT emergence, our results clarify this prevailing notion for the first time. Furthermore, we also presented evidence in Appendix E.6 that practical networks such as VGG can indeed exhibit HT ESDs with large enough $\eta$ and without gradient noise.

---

> ### Author Response · Authors · 2024-11-16
>
> >> (c) The theoritical analysis does not capture how "BULK + spike" transition into a heavy tail phenomenon
>
> We acknowledge that our theoretical results are currently applicable to the bulk+spike ESD after one step of Adam and do not extend to multiple steps. We discuss this limitation in Appendix G. In particular, since we use large learning rates, our setup is not amenable to continuous time approximation techniques to potentially study the ESD evolution. Furthermore, the Gaussian equivalence of the weight matrices does not hold in this regime which allowed Ba et.al to study the role of finite training steps on the weight matrix and generalization. To this end, we kindly note that establishing theoretical results to rigorously characterize the dynamics of the ESD evolution requires an independent analysis of its own (especially in this large lr regime).
>
> **We have fixed the minor notation issues in the revision. Thanks for pointing them out.**
>
> **References:**
>
> [1] Martin, Charles H., and Michael W. Mahoney. "Implicit self-regularization in deep neural networks: Evidence from random matrix theory and implications for learning." Journal of Machine Learning Research 22.165 (2021): 1-73.
>
> [2] Umut Simsekli, Levent Sagun, and Mert Gurbuzbalaban. A tail-index analysis of stochastic gradient
> noise in deep neural networks. In International Conference on Machine Learning, pp. 5827–5837. PMLR, 2019.
>
> [3] Wang, Zhichao, et al. "Spectral evolution and invariance in linear-width neural networks." Advances in Neural Information Processing Systems 36 (2024).
>
> [4] Ba, Jimmy, et al. "High-dimensional asymptotics of feature learning: How one gradient step improves the representation." Advances in Neural Information Processing Systems 35 (2022): 37932-37946.

---

> > ### Author Response · Authors · 2024-11-24
> > **Requesting feedback on our response**
> >
> > Gentle ping to request feedback on our response. We are happy to any further questions as well.
> >
> > - Authors

---

> > > ### Comment · Reviewer_uys5 · 2024-11-25
> > > **Reply to the authors**
> > >
> > > I appreciate the authors’ detailed reply. However, I remain unconvinced on the following points:
> > > (a) Whether this setup effectively captures the correlation between the HT spectrum and generalization, and
> > > (b) Even if this is the right setup, whether the paper's theoretical contributions sufficiently address this correlation.
> > >
> > > Below, I outline my concerns:
> > >
> > > **Wang et al. demonstrated that mini-batch training with Adam can facilitate the emergence of HT-ESD but did not provide theoretical results to explain such emergence.**
> > > Similarly, I believe this paper does not theoretically explain the emergence of HT-ESD and instead focuses on Bulk + Spike analysis. Hence, the paper only partially answers this question through Bulk + spike after single step.
> > >
> > > After reading the rebuttal, here is my summary of the authors' responses and claims. Please let me know if I have misunderstood anything:
> > >
> > > 1. The paper demonstrates that heavy tails in the ESD can arise without gradient noise.
> > > 2. For single-index models, a spike occurs after a step of Adam for certain ranges of $\eta$.
> > >
> > > I agree that these are interesting contributions of the paper. However, I do not entirely agree with some of the other claims made by the authors:
> > >
> > > 3.  I disagree with the assertion that the paper _analyzes the role of learning rates or techniques such as weight normalization and learning rate schedules_. Studying a single-step update does not fully characterize the role of learning rate or their schedules.
> > >
> > > **Bulk + Spike vs. HT:**
> > >
> > > From the authors’ response and upon revisiting the paper, it appears that the HT-ESD is a byproduct of the spike, with the paper primarily focusing on the formation of the spike. Based on this, it seems that **Bulk + Spike is more directly linked to generalization than the HT spectrum.** This makes me question the setup to understand the correlation between HT spectrum and generalization, since the formation of spike seems to be the more important phenomenon that makes HT irrelevant.
> > >
> > > Moreover, the paper does not address how the spike evolves into heavy tails over time. Consequently, I disagree with the claim that the paper *comprehensively studies the HT spectrum, its influence on generalization*. To summarize, while I agree that the paper investigates the evolution of the spike after one step of Adam, I do not believe it makes any significant theoretical contribution to understanding Heavy Tail ESD or its correlation with generalization.

---

> > > > ### Author Response · Authors · 2024-11-30
> > > > **Requesting feedback on the clarifications**
> > > >
> > > > Dear Reviewer uys5,
> > > >
> > > > As the discussion deadline is approaching, we kindly request feedback on our response and clarifications. We are happy to answer any further questions.
> > > >
> > > > Thanks again for your time.

---

> ### Author Response · Authors · 2024-11-25
> **Clarifying the misunderstandings**
>
> We appreciate the reviewer's response and for engaging in a discussion. We clarify some misunderstandings in the reviewer's response below. Please note that:
>
> 1. Our theoretical contributions pertain to the Bulk+Spike phase after one step of Adam, and we do not claim to provide a theory for the complete evolution dynamics into an HT ESD. We kindly note that these claims have been stated in the introduction and we discuss the limitations of our work in Appendix G. On a related note, we emphasize that our empirical analysis of the HT ESD emergence (Section 5) lays the groundwork for future efforts to address our limitations. We hope that this clarifies our claims on the theoretical aspects of our work (See also line 76 in the paper).
>
> 2. **Regarding the reviewer's comment on learning rates, weight normalization and lr schedules:** We have presented extensive experimental results in Appendix E.4 to study the role of weight normalization and lr schedules with multiple optimizer steps and their impact on ESD and generalization in our setup (see also the paragraph at line 465 in main paper). We re-emphasize that our analysis is empirical for these aspects of HT-ESD emergence and hope that this clarifies the misunderstanding regarding the claims.
>
> 3. The reviewer's point on the importance of the `spike' for the emergence of HT ESDs is essentially the crux of our paper. However, we would like to clarify a misunderstanding. **Please note that not all Bulk+spike ESDs lead to good generalization**. More importantly, for very large learning rates (for ex: $\eta>10$), we can observe from Figure 2 that the spike emerges but the generalization of the NN is poor (Figure 3). Figure 5 shows that the generalization continues to be poor even after multiple steps. This behavior can be understood by the shape of the HT ESDs whose PL Alpha fit is $< 2$ and indicates extremely heavy-tailed spectra. Thus, the shape of the HT ESD also acts as an indicator of whether the network can generalize well or not. This makes it an important aspect to study beyond the spike after the first step. We hope that this clarifies any confusion regarding its relevance (See also Martin et,al [1], (reference above) who observed such behavior in deeper NNs).
>
> Finally, we re-emphasize point 1 above that **we do not claim** to provide a full theory for the evolution of ESD into HT ESD (line 76 in paper). We discuss these aspects in Appendix G. We note that this work aims to theoretically characterize the Bulk+spike phase with Adam (which is new to the literature) and present empirical evidence to show that this setup can indeed be used to study HT-ESDs after multiple optimizer steps. This aspect has also been discussed with other reviewers below.
>
> We kindly request the reviewer to reconsider after these clarifications, and we are happy to take your suggestions on updating the presentation or clarify any further questions. Thank you.

---

### Meta-Review · Area_Chair_g5KQ · 2024-12-10

**Metareview:**

The paper explores the emergence of heavy-tailed in the empirical spectral distributions of the neural network weights during training. It focuses on the transition from a bulk + spike to heavy tails. While the paper presents some interesting insights, such as Adam requiring smaller step sizes than GD for spike emergence, the reviews highlight mixed opinions.

### **Strengths**

- Novel exploration of HT-ESD emergence in Adam, extending prior work on GD.
- Interesting empirical analysis
- Insights into the spectral properties of weight matrices and their role in generalization.


### **Weaknesses**
- The theoretical contributions are limited to the bulk + spike phase after one step
- The experiments are restricted to toy models, raising concerns about generalizability.

### **Decision**
Two reviewers (bo67, xTm5) support acceptance while two (uys5, piS8) recommend rejection, citing insufficient depth and limited applicability. Overall, the paper offers valuable early-stage insights but lacks the theoretical and empirical depth to comprehensively address HT-ESD and its connection to generalization. I believe this requires additional work.

**Additional Comments On Reviewer Discussion:**

Two reviewers (bo67, xTm5) support acceptance while two (uys5, piS8) recommend rejection, citing insufficient depth and limited applicability. The discussion period did not convince the two reviewers in favor of rejection.

---

### Decision · Program_Chairs · 2025-01-22

Reject